# Magnetic Field Analysis and Performance Optimization of Dual-Rotor Hybrid Excitation Generator for Automobile

Shilong Yan [1], Xueyi Zhang [1,*], Jun Zhang [2], Yufeng Zhang [1], Mingjun Xu [1], Ting Gao [1] and Sizhan Hua [1]

1    School of Transportation and Vehicle Engineering, Shandong University of Technology, Zibo 255049, China
2    Shandong Tangjun Ouling Automobile Manufacturing Co., Ltd., Zibo 255185, China
*    Correspondence: zhangxueyi@sdut.edu.cn

**Abstract:** Aiming at the current problems of low excitation efficiency and poor reliability of single-rotor hybrid excitation generators, the large axial length of dual-rotor structure, and difficulty in magnetic field analysis, a new type of the dual-rotor hybrid excitation generator topology with high power density is proposed, with two rotors side-by-side coaxial, sharing a set of armature windings, and the magnetic fields do not interfere with each other, so the magnetic field analysis and optimization of the two rotors can be carried out separately. The magnetic density distribution of the new permanent magnet (PM) claw pole rotor is analyzed by the joint application of the equivalent magnetic circuit method and the equivalent magnetic network method, which ensures the simplicity of calculation and improves the calculation accuracy. The multi-objective optimization of the key structural parameters is carried out based on the Latin hypercube sampling–Pareto frontier solution method. The subdomain method is improved by segmented equivalence, the unique solution of the salient-pole rotor magnetic field is obtained, and the multi-objective optimization of the salient-pole rotor is used by the particle swarm algorithm. The trial prototype was experimental, and the results showed that the output characteristics of the optimized hybrid excitation generator were significantly improved, and the overall performance of the generator was improved.

**Keywords:** hybrid excitation generator; dual-rotor; magnetic density of the air gap; multi-objective optimization





## 1. Introduction

The generator is one of the key components of vehicles such as internal combustion engine vehicles, new energy hybrid vehicles, military vehicles, and so on. In recent years, people's growing demand for a better life has also determined that the requirements for automobile intelligence are getting higher and higher, and the "intelligence" of automobiles is inseparable from the support of various auxiliary electrical equipment, and the corresponding increase in electricity consumption has put forward new requirements for automobile generators. The traditional rare earth PM generator has the advantages of small size, lightweight, high efficiency, superior performance, and strong overload capacity. However, the magnetic characteristics of the PM after magnetization in rare earth PM generators are difficult to adjust, so when the load and speed of the generator change, the output voltage of the generator is difficult to stabilize. To solve the problem of difficult voltage stabilization of ordinary rare earth PM generators, compound excitation technology is proposed, and the compound excitation technology is applied to PM synchronous generators so that the output voltage of PM generators can be adjusted [1–5]. Compound excitation synchronous generator inherited many characteristics of the PM synchronous generator, the structure is different from the electric excitation synchronous generator and PM synchronous generator, the two magnetic potential sources in the compound excitation synchronous generator exist at the same time, and the excitation is adjustable, it integrates

the electric excitation synchronous generator to regulate the convenience and PM synchronous generator has high efficiency, torque/quality ratio, and other advantages, and has a wide range of dissemination and application value.

At present, many scholars in the world have conducted in-depth research on the compound excitation generator. W. Yan and S. Jin proposed a new type of double-stator doubly-fed compound excitation generator [6,7]. This type of generator is generally composed of an internal and external double-stator and hollow cup rotor, and the excitation windings are embedded on the teeth of the inner stator to achieve brushless, the stator electric-magnetic field is synthesized in parallel with the permanent magnetic field in the air gap through the rotor core to form a compound excitation mode. Scholar H. Wang proposed a new type of hybrid excitation permanent magnet synchronous generator [8], the permanent magnetic field of the motor is synthesized in parallel with the electric magnetic field, and the stator windings are composed of armature windings and stator DC excitation windings, and the two are layered winding, the permanent magnet is fixed on the outer circle of the rotor. Scholar T.A. Lipo proposed a new type of compound excitation synchronous generator [9,10], the generator rotor is composed of a 4-pole PM pole and a 2-pole electric excitation pole, and the change in the direction of the excitation current can change the number of poles of the generator, thereby forming a variable pole structure. S. Hlioui proposed a brushless compound excitation generator with field windings on the stator [11,12], the generator adds additional magnetic guide rings on both sides of the rotor so that the magnetic field generated by the field winding passes through the end cover and the magnetic guide ring, and then is in parallel with the PM magnetic field. Scholar B. Gaussens proposed a flux switching generator [13], the generator rotor is a salient-pole rotor, there is neither PM nor electric field winding on the rotor, the structure is simple, suitable for the working conditions with higher speed requirements, the stator is "E" type stator core, the electric field winding is embedded in the middle tooth of the "E" type stator core, and the rectangular PM is installed in the radial rectangular groove between the two adjacent "E" type stator cores.

To reduce the axial length of the generator, scholars have proposed that the compound excitation generator is mostly a single-rotor structure, and the introduction of the electro-excitation magnetic field has many limitations and defects such as low excitation efficiency and large magnetic circuit magnetoresistance [14–16].

The dual-rotor structure can effectively solve the problem of the introduction of electric excitation, and the magnetic field generated by the electric excitation winding directly enters the stator and then generates the induced electromotive force (EMF), which has little impact on the PM magnetic field. The dual rotor makes the combination of magnetic fields more flexible, and different rotors can be selected according to different engineering applications to meet the needs. In addition, under the same axial length, compared with the single-rotor structure, the dual-rotor structure can effectively reduce the eddy current loss of the generator due to its segmented treatment, which is of great significance to improving the power density and efficiency of the generator.

The work arrangement of this paper is as follows. The second section introduces the electromagnetic field analysis of the dual-rotor hybrid excitation generator and establishes the magnetic density analytical model of the claw pole part and the convex pole part. The third section introduces the optimization method used, optimizes the design variables, and makes a comparative analysis of the optimization results. The fourth part summarizes the work.

## 2. Analysis of Electromagnetic Field of Dual-Rotor Hybrid Excitation Generator

Combined with the specific engineering application field, a new compound excitation generator structure is proposed, so that the generator has a PM claw pole rotor and a parallel magnetic circuit salient-pole rotor, the two are parallel coaxial, sharing an armature winding, the magnetic field generated is synthesized in the air gap, and the structure shortens the axial length of the two rotors, as far as possible to reduce the demand for axial

space of the dual-rotor structure. The overall structure of the new compound excitation generator is shown in Figure 1.

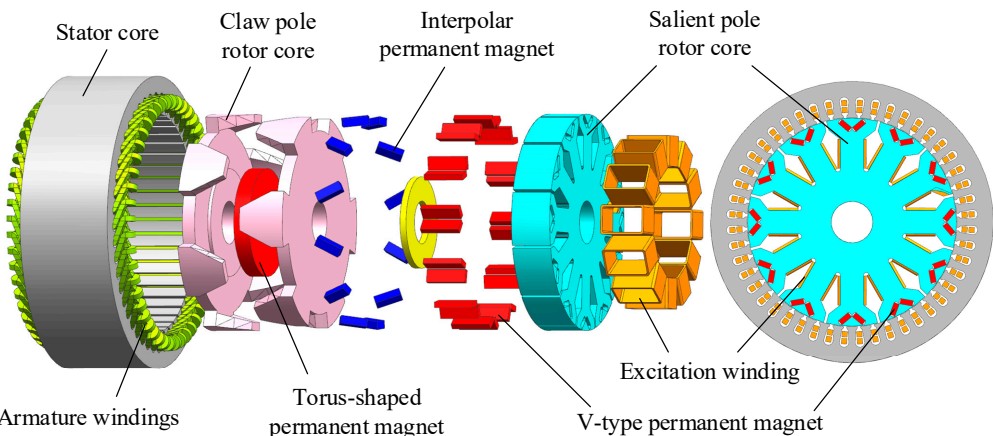

**Figure 1.** The structure of the new compound excitation generator.

There are many forms of built-in permanent magnets in the pole shoes. The more common built-in permanent magnets include "-" type, "V" type, "W" type, etc. The magnetic density produced by several forms of permanent magnets is analyzed by finite element software. The air gap magnetic density of different built-in permanent magnet salient pole rotors is shown in Figure 2.

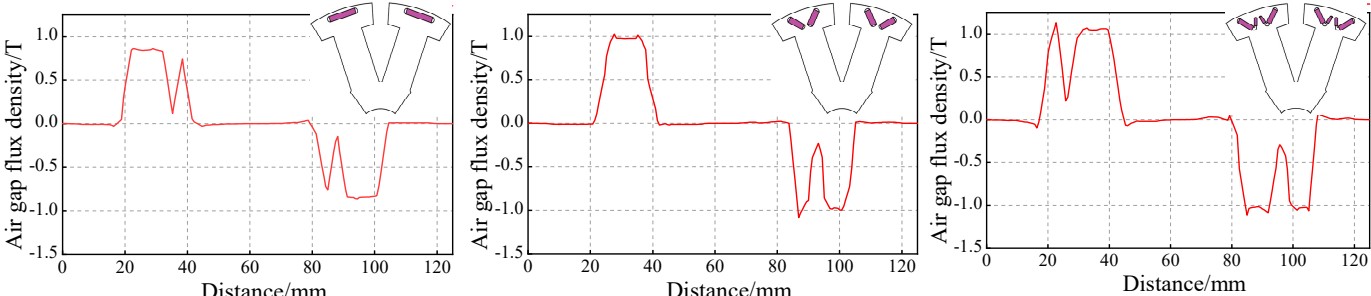

**Figure 2.** Air-gap flux density of different built-in permanent magnet salient pole rotors.

It can be seen from Figure 2 that although the "-" type permanent magnet has a simple structure and convenient placement, the air-gap magnetic density peak generated by it is only 0.83 T, while that generated by the "V" type permanent magnet and the "W" type permanent magnet are 1.03 T and 1.12 T, respectively. Obviously, the latter two have good magnetic agglomeration effect. In addition, considering the limited space of the salient rotor, the "V" type permanent magnet is more space-saving, and the increase of the air-gap magnetic density peak generated by the "W" type permanent magnet is limited compared with the "V" type permanent magnet. Therefore, the "V" type permanent magnet is built into the pole boots.

Since in a dual-rotor generator, the magnetic field interference between the two rotors is small, it can be considered as two generators, respectively. Therefore, the magnetic field analysis of the compound excitation generator can be divided into two parts: the magnetic field analysis of the new PM claw pole based on the PM claw pole generator, and the magnetic field analysis of the new salient-pole rotor based on the salient-pole generator.

### 2.1. Subsection Analytical Model of Magnetic Flux Density of Claw Pole Generator

In the new compound excitation generator structure, to improve the air gap magnetic density of the claw pole rotor and reduce the inter-electrode flux leakage, a tangential

magnetized interpolate PM is added between the claw teeth. When the magnetic field of the claw pole generator is analyzed, the corresponding mathematical model will also change greatly.

The magnetic circuit analysis of the claw pole generator mainly applies the equivalent magnetic circuit method and makes the following assumptions [17]:

Magnetic saturation of the stator and rotor cores is not taken into account. It is assumed that the demagnetization curve of the PM is a straight line, the recovery line of the PM and the demagnetization curve coincide, and the PM magnetization is uniform. The influence of the armature winding end is not considered.

### 2.1.1. Magnetic Flux Path Analysis of Claw Pole Generator

The main magnetic flux path of the claw pole generator is relatively simple, but after adding the inter-pole PM, the generator adds a new main magnetic flux route. A schematic diagram of the main magnetic circuit of the claw pole generator is shown in Figure 3.

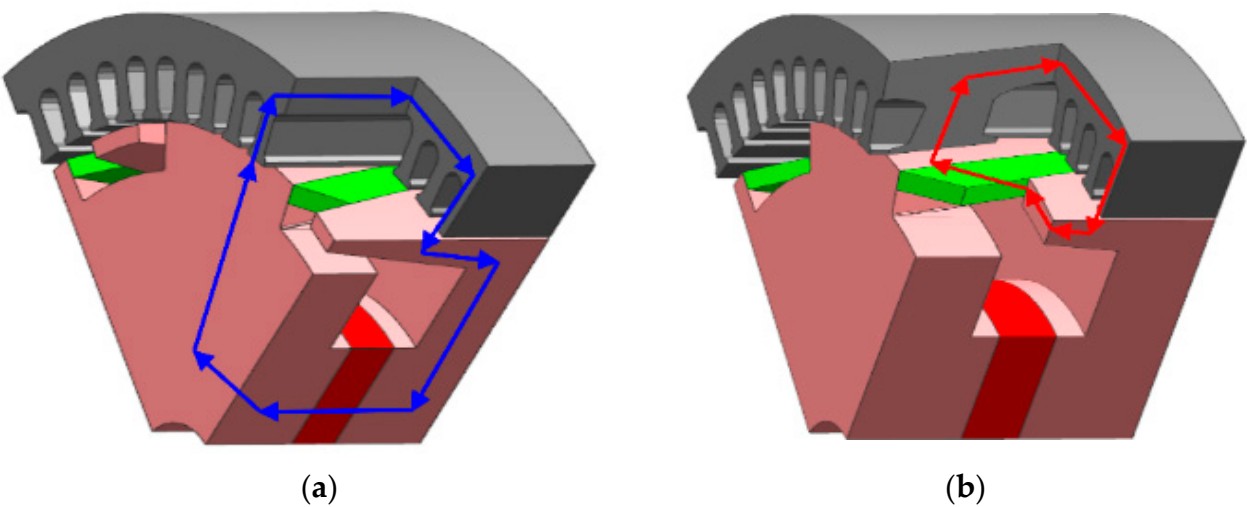

**Figure 3.** Schematic diagram of the main magnetic circuit of claw pole generator. (**a**) The first main magnetic circuit; (**b**) the second main magnetic circuit.

As can be seen from Figure 3, the main magnetic flux path 1 of the generator is toroidal PM N pole—claw pole rotor front claw pole yoke, flange, tooth part—main air gap—stator core—main air gap—claw pole rotor rear claw pole tooth part, flange, magnetic yoke—toroidal PM S pole. The main magnetic flux path 2 of the generator is inter-pole PM N pole—front claw pole tooth part of the main air gap—stator core—main air gap—rear claw pole tooth part—inter-pole PM S pole.

The addition of inter-pole PMs adds a new main magnetic circuit, which cancels the leakage of magnetism between the claw poles but also increases the new leakage of magnetic flux, and the leakage flux of the new generator is mainly divided into leakage magnetism inside the claw pole, leakage between the claw pole poles, and leakage magnetic flux outside the claw pole. Due to the addition of the PM between the poles, the magnetic flux leakage between claws is effectively weakened and can be ignored, while the external magnetic resistance of claws is large and the magnetic flux leakage can also be ignored. The schematic diagram of the leakage flux path of the claw pole generator is shown in Figure 4.

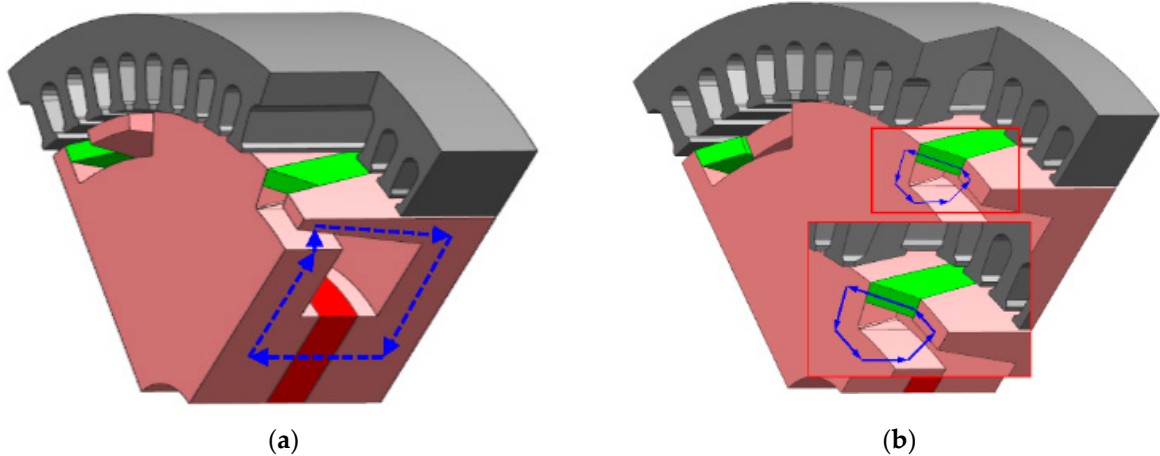

**Figure 4.** Schematic diagram of the leakage flux path of claw pole generator. (**a**) The first main magnetic circuit; (**b**) the second main magnetic circuit.

As can be seen from Figure 4, the main leakage flux path of the rear claw pole of the inter-pole PM can be divided into two parts, the first is the leakage flux generated by the toroidal permanent magnet: the N pole of the toroidal permanent magnet—the front claw pole yoke, the flange—the air gap inside the claw pole—the rear claw pole flange, the magnetic yoke—the toroidal permanent magnet S pole. The second is the leakage flux generated by the inter-pole PM: the inter-pole permanent magnet N pole —the front paw pole tooth part—the internal air gap of the claw pole—the rear claw pole tooth part—the inter-pole permanent magnet S pole.

### 2.1.2. Magnetic Flux Analysis of Claw Pole Generator

As far as the synchronous generator is concerned, the sinusoidal characteristics of the air gap magnetic density waveform will directly affect the performance of the generator, and the higher the waveform distortion rate of the air gap flux density waveform is, the higher the waveform distortion rate of induced EMF waveform of the motor will be. The fundamental amplitude of the air gap magnetic density directly affects the output voltage of the generator. Therefore, accurate analysis and calculation of generator air gap magnetic density is of great significance for the theoretical analysis of the generator. The traditional equivalent magnetic circuit method is not suitable for analyzing three-dimensional magnetic fields, so this paper uses the equivalent magnetic circuit method and the equivalent magnetic network method to calculate the air gap magnetic density of the claw pole generator, which reduces the difficulty of calculation and improves the calculation accuracy.

Based on the analysis of the main flux path and the leakage flux, the equivalent magnetic circuit diagram of the claw pole generator can be obtained. When an inter-pole permanent magnet is added, the magnetic field generated by the inter-pole PM is paralleled with the magnetic field generated by the toroidal PM. The equivalent magnetic circuit diagram of the claw pole generator containing the inter-pole PM is shown in Figure 5.

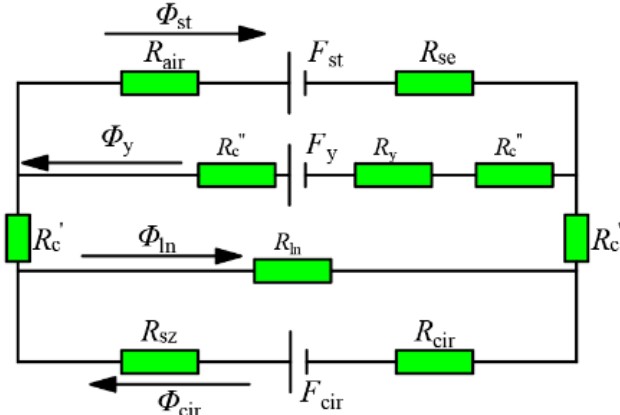

**Figure 5.** Equivalent magnetic circuit diagram of claw pole generator with inter-pole PM.

$\Phi_{st}$ is the magnetic flux through the air gap, $\Phi_{lm}$ is the external flux leakage of the rotor, $\Phi_{ln}$ is the internal flux leakage of the rotor, $\Phi_{cir}$ is the magnetic flux generated by a toroidal PM, $\Phi_y$ is the magnetic flux generated by the PM between the poles; $F_{st}$ is the stator magnetomotive force, $F_{cir}$ is the magnetomotive force generated by a toroidal PM, $F_y$ is the magnetomotive force generated by the PM between the poles; $R_{air}$ is the air gap magnetoresistance, $R_{se}$ is the stator magnetoresistance, $R_{lm}$ is the external leakage magnetoresistance of the rotor, $R_c$ is the rotor tooth magnetoresistance, $R_y$ is the magnetoresistance of the PM between the poles, $R_{ln}$ is the leakage magnetoresistance inside the rotor, $R_{sz}$ is the magnetoresistance of the claw rotor except for the tooth part, $R_{cir}$ is the magnetoresistance of the toroidal PM, $R_c'$ is the axial magnetoresistance of the rotor teeth, $R_c''$ is the tangential magnetoresistance of the rotor teeth.

After adding the inter-pole PM, the leakage of the magnet between the claw pole poles is suppressed, which can be ignored, and the magnetic circuit relationship of the claw pole generator can be obtained:

$$\begin{cases} \Phi_{st} = \Phi_{cir} - \Phi_{ln} + \Phi_y \\ F_{cir} - \Phi_{cir}(R_{sz} + R_{cir}) = \Phi_{ln}R_{ln} \\ \Phi_{ln}R_{ln} + 2(\Phi_{cir} - \Phi_{ln})R_c' = F_y - \Phi_y(2R_c'' + R_y) \\ F_y - \Phi_y(2R_c'' + R_y) = F_{st} + \Phi_{st}(R_{air} + R_{se}) \end{cases} \tag{1}$$

The air gap magnetic flux of the generator can be obtained:

$$\Phi_{st} = \frac{1}{S_1}\left[ \begin{array}{c} \frac{F_y R_{ln} - F_{cir}R_{ln} + F_{cir}R_y}{R_{ln}\left(2R_c'' + R_y\right)} \\ + S_2 \times \frac{F_{st}R_{ln} - F_{cir}R_{ln} + 2F_{cir}R_c'}{2R_c'(R_{ln} + R_{sz} + R_{cir}) - R_{ln}(R_{sz} + R_{cir})} \end{array} \right] \tag{2}$$

Among them, the constants $S_1$ and $S_2$ are:

$$S_1 = 1 - \frac{R_{ln}(R_{air} + R_{se})\left(2R_c'' + R_y\right)(R_{ln} + R_{sz} + R_{cir})}{R_{ln}\left(2R_c'' + R_y\right)[2R_c'(R_{ln} + R_{sz} + R_{cir}) - R_{sz}R_{ln} - R_{cir}R_{ln}]}$$
$$+ \frac{R_{ln}(R_{air} + R_{se})[R_{ln}(R_{sz} + R_{cir} - 2R_c') - 2R_c'(R_{sz} + R_{cir})]}{R_{ln}\left(2R_c'' + R_y\right)[2R_c'(R_{ln} + R_{sz} + R_{cir}) - R_{sz}R_{ln} - R_{cir}R_{ln}]} \tag{3}$$

$$S_2 = \frac{\left(2R_c'' + R_y\right)(R_{ln} + R_{sz} + R_{cir}) + R_{ln}(R_{sz} + R_{cir} - 2R_c') - 2R_c'(R_{sz} + R_{cir})}{R_{ln}\left(2R_c'' + R_y\right)} \tag{4}$$

where:

$$\begin{cases} F_{cir} = H_c h_{cir} \\ F_y = H_c h_{jyc} \\ F_{st} = 4.44K_e f N_{st}\Phi_{st} \end{cases} \tag{5}$$

where $H_c$ is the PM coercivity, $h_{cir}$ is the magnetization thickness of the toroidal PM, $h_{jyc}$ is the magnetization thickness of the inter-pole PM, $K_e$ is the magnetic flux coefficient, $f$ is the generator frequency, and $N_{st}$ is the number of turns per slot for the armature windings.

Then, the air gap magnetic density generated by it can be expressed as:

$$B_g = \frac{\Phi_{st}}{A_g} = \frac{180\Phi_{st}}{\beta_z \pi r_g L_{ef1}} \tag{6}$$

where $\beta_z$ is the mechanical angle occupied by each pole of the claw pole rotor, $r_g$ is the air gap radius, and $L_{ef1}$ is the axial length of the claw pole rotor. Then, the expression of the generator induced EMF can be expressed as:

$$E = B_g L_{ef1} v \frac{N_{st}}{a} = \frac{\varphi}{L_{ef1}\tau} L_{ef1} v \frac{N_{st}}{a} = \frac{1}{a} \frac{N_{st}\Phi_{st}n_N}{60} \tag{7}$$

Namely:

$$E = \frac{N_{st}n_N}{60a} \bullet \frac{1}{S_1} \left[ \begin{array}{c} \frac{F_y R_{ln} - F_{cir} R_{ln} + F_{cir} R_y}{R_{ln}\left(2R_c'' + R_y\right)} \\ +S_2 \bullet \frac{F_{st} R_{ln} - F_{cir} R_{ln} + 2F_{cir} R_c'}{2R_c'(R_{ln}+R_{sz}+R_{cir}) - R_{ln}(R_{sz}+R_{cir})} \end{array} \right] \tag{8}$$

where $n_N$ is the rated speed of the generator, and $a$ is the logarithm of the parallel branches.

### 2.1.3. Calculation of the Magnetoresistance of Each Part

The calculation accuracy of the magnetoresistance of each part of the generator determines the accuracy of the mathematical model of the air gap magnetic density, because the claw pole generator has a complex three-dimensional magnetic field, the use of the traditional equivalent magnetic circuit method cannot meet the calculation requirements of the magnetic field of the claw pole generator, so the equivalent magnetic network method is used to calculate the magnetoresistance of the components of the claw pole generator.

The principle of the equivalent magnetic network method is to differentiate the generator into several units with uniform magnetic flux distribution, connect each unit according to its actual position in the generator, and finally form a magnetic network [18]. The equivalent magnetic network model of the traditional claw pole generator is simple in structure, and the model only considers the axial magnetic conductivity of the claw tooth part of the generator, which does not apply to the claw pole generator with an inter-pole PM proposed in this paper.

Since the direction of the inter-pole PM is tangentially magnetized, it is necessary to consider its tangential magnetic conductivity when considering the tooth part of the claw pole rotor, and due to the addition of the inter-pole PM, the generator adds a new main magnetic flux and reduces the leakage of some inter-pole magnets, so a more accurate and improved equivalent magnetic network model needs to be established. According to the magnetic flux distribution in the claw pole generator, the generator is divided into rotor tooth part, rotor yoke part, magnetic conductive ring, stator tooth part, stator yoke part, PM, armature winding, and air gap, etc. For convenience, an equivalent magnetic network model of a pair of poles of the generator under radial cross-section is given. The improved equivalent magnetic network model is shown in Figure 6.

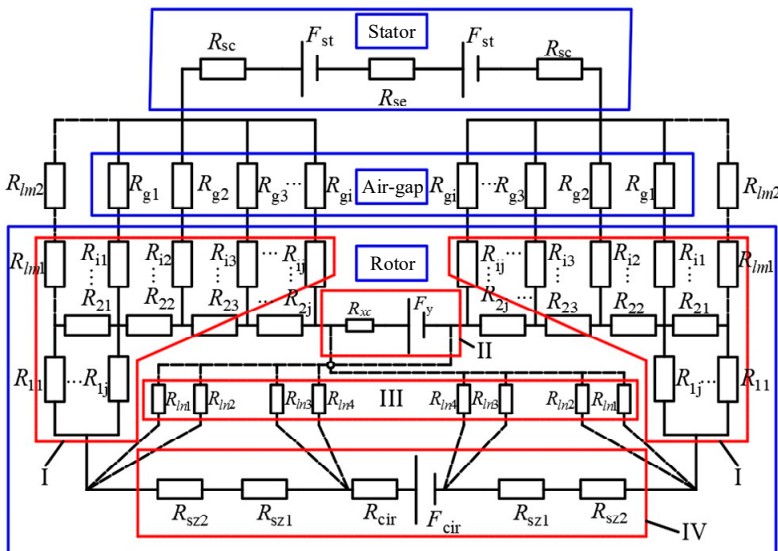

**Figure 6.** Improved equivalent magnetic network model.

In Figure 6, region I represents the magnetic network model of the rotor tooth and yoke, region II represents the equivalent expression of the inter-pole PM, region III is the leakage magnet inside the claw pole, and region IV is the equivalent expression of the toroidal PM and the magnetic ring part. The solid lines in the figure represent the main flux path, and the dashed lines represent the leakage flux path.

In the tangential direction, the radial section of the claw pole rotor changes with the change of angle $\alpha$. Its cross-sectional area varies symmetrically along the central axis of the rotor teeth, so only half of the rotor teeth need to be considered. It is defined that when the radial section passes through the central axis of the rotor tooth, $\alpha = 0$, the cross-sectional area of the rotor tooth decreases with the increase of $\alpha$, and the number of elements divided by the rotor tooth also decreases. The subdivision unit is shown in Figure 7.

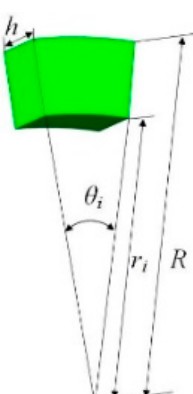

**Figure 7.** The subdivision unit.

In Figure 7, $h_i$, $r_j$ represents the axial distance and inner diameter of the axial $i$th and tangential $j$th of the subdivision unit, $\theta_{ij}$ represents the center angle of the subdivision unit, and $R$ is the outer radius of the generator claw pole rotor. The axial magnetoresistance and tangential magnetoresistance of each subdivision unit of the claw pole rotor are respectively [19]:

$$R'_{ij} = \frac{l}{\mu s} = \frac{1}{\mu_c} \frac{1}{\frac{1}{2}\theta_i(r_c^2 - r_j^2)} = \frac{2h_i}{\mu_c \theta_{ij}(r_c^2 - r_j^2)} \tag{9}$$

$$R''_{ij} = \frac{l}{\mu s} = \frac{1}{\mu_c \int_{r_j}^{r_c} \frac{h_i}{r\theta_{ij}} dr} = \frac{\theta_{ij}}{\mu_c h_i (\ln r_c - \ln r_j)} \quad (10)$$

where $\mu_c$ is the relative permeability of the claw pole core, $r_c$ is the radius of a subdivision unit at the claw tooth. Then, the tooth magnetoresistance $R_c$ of the claw pole rotor can be expressed as:

$$R'_c = \begin{cases} \frac{2l_{cc}}{\mu_c \beta_{cj}(R^2 - (R - h_{cj})^2)}, & -\frac{\beta_{cj}}{2} < \alpha < \frac{\beta_{cj}}{2}, R - h_{cj} < r < R \\ \int_{R - h_{cg}}^{R - h_{cj}} \left[ \sum_{i=1}^{n} \frac{2h_i}{\mu_c \beta_{cj}((R - h_{cj})^2 - r^2)} \right] dr, & -\frac{\beta_{cj}}{2} < \alpha < \frac{\beta_{cj}}{2}, R - h_{cg} < r < R - h_{cj} \\ \int_{\frac{\beta_{cg} - \beta_{cj}}{2}}^{\frac{\beta_{cg}}{2}} \left[ \sum_{i=1}^{n} \frac{2h_i}{\mu_c \alpha(R^2 - (R - h_{cj})^2)} \right] d\alpha, & \frac{\beta_{cj}}{2} < |\alpha| < \frac{\beta_{cg}}{2}, R - h_{cj} < r < R \\ \int_{R - h_{cg}}^{R - h_{cj}} \left[ \int_{\frac{\beta_{cg} - h_{cj}}{2}}^{\frac{\beta_{cg}}{2}} \left[ \sum_{i=1}^{n} \frac{2h_i}{\mu_c \alpha((R - h_{cj})^2 - r^2)} \right] d\alpha \right] dr, & \frac{\beta_{cj}}{2} < |\alpha| < \frac{\beta_{cg}}{2}, R - h_{cg} < r < R - h_{cj} \end{cases} \quad (11)$$

$$R''_c = \begin{cases} \frac{\beta_{cj}}{\mu_c l_{cc}[\ln R - \ln(R - h_{cj})]}, & -\frac{\beta_{cj}}{2} < \alpha < \frac{\beta_{cj}}{2}, R - h_{cj} < r < R \\ \int_{R - h_{cg}}^{R - h_{cj}} \left[ \sum_{j=1}^{n} \frac{\beta_{cj}}{\mu_c l(r)[\ln(R - h_{cj}) - \ln r_j]} \right] dr, & -\frac{\beta_{cj}}{2} < \alpha < \frac{\beta_{cj}}{2}, R - h_{cg} < r < R - h_{cj} \\ \int_{\frac{\beta_{cg} - \beta_{cj}}{2}}^{\frac{\beta_{cg}}{2}} \left[ \frac{\beta_{cg} - \beta_{cj}}{2\mu_c l(\alpha)[\ln R - \ln(R - h_{cj})]} \right] d\alpha, & \frac{\beta_{cj}}{2} < |\alpha| < \frac{\beta_{cg}}{2}, R - h_{cj} < r < R \\ \int_{R - h_{cg}}^{R - h_{cj}} \left[ \int_{\frac{\beta_{cg} - h_{cj}}{2}}^{\frac{\beta_{cg}}{2}} \left[ \sum_{j=1}^{n} \frac{\beta_{cg} - \beta_{cj}}{\mu_c l(\alpha < r)[\ln(R - h_{cj}) - \ln r_j]} \right] d\alpha \right] dr, & \frac{\beta_{cj}}{2} < |\alpha| < \frac{\beta_{cg}}{2}, R - h_{cg} < r < R - h_{cj} \end{cases} \quad (12)$$

where $l_{cc}$ is the axial length of the claw pole tooth, $\beta_{cj}$ is the mechanical angle occupied by the tip of the claw pole tooth, $\beta_{cg}$ is the mechanical angle occupied by the root of the claw pole tooth, $h_{cj}$ is the claw tooth tip thickness, $h_{cg}$ is the claw tooth root thickness.

To simplify the calculation process, the parts with uniform magnetoresistance and regular shape need not be solved separately, then the magnetoresistance $R_{sz}$ except for the tooth part of the claw pole core is the sum of the magnetoresistance of the claw pole flange and the yoke part, and the two parts of the magnetoresistance are expressed as follows:

$$\begin{cases} R_{flp} = \frac{1}{G_{flp}} = \frac{l_{flp}}{\mu_c A_{flp}} \\ R_{ce} = \frac{1}{G_{ce}} = \frac{l_{ce}}{\mu_c A_{ce}} \end{cases} \quad (13)$$

Thereinto:

$$\begin{cases} l_{flp} = \frac{1}{2} D_{i1} - 2\delta - 2h_{cg} \\ A_{flp} = \frac{\pi r_x h_{flp} \beta_{cg}}{180} \\ A_{ce} = \pi (r_{cir}^2 - r_{i3}^2) \end{cases} \quad (14)$$

where $R_{flp}$ and $R_{ce}$ are the flange magnetoresistance and the yoke magnetoresistance, $l_{flp}$ and $l_{ce}$ are the length of the magnetic circuit in the flange and the length of the magnetic circuit in the yoke, $A_{flp}$ and $A_{ce}$ are the magnetic circuit area in the flange and the magnetic circuit area in the yoke, $h_{flp}$ is the axial thickness of the flange, $r_x$ is the radius of the circle corresponding to any point in the flange, $r_{cir}$ is the outer diameter of the toroidal PM, $r_{i3}$ is the inner radius of the claw pole rotor. If the magnetoresistance in the air is uniformly distributed, the main air gap magnetoresistance can be expressed as follows:

$$R_{air} = \frac{1}{G_{air}} = \frac{p\delta}{\mu_0 \pi D_{i1} L_p} \quad (15)$$

Due to the small stator slot shoulder, the magnetoresistance $R_{st}$ of the stator part can be divided into two parts, the stator tooth and the stator yoke, and for the convenience of solving, the two parts are regarded as regular shapes, and its expression is as follows:

$$\begin{cases} R_{stc} = \frac{1}{G_{stc}} = \frac{l_{stc}}{\mu_c A_{stc}} \\ R_{ste} = \frac{1}{G_{ste}} = \frac{l_{ste}}{\mu_c A_{ste}} \end{cases} \tag{16}$$

where $R_{stc}$ and $R_{ste}$ are the magnetoresistance of the stator tooth and the stator yoke, $l_{stc}$ and $l_{ste}$ are the lengths of the stator tooth and stator yoke circuit, $A_{stc}$ and $A_{ste}$ are the magnetic circuit areas in the stator tooth part and the stator yoke part.

*2.2. Analytical Model of Air Gap Magnetic Density of Salient Pole Generator*

The traditional subdomain method has stricter requirements for the distribution of the midfield of each subdomain [20]. To meet the solution adjustment of Poisson's equation and Laplace's equation, the Kirchhoff's law is used to equate the salient-pole generator as a surface-mount PM generator structure, with the magnetic vector position as the solution variable, and the magnetic vector position of each subdomain is solved by the separation variable method, and each order harmonic coefficient is solved according to the boundary conditions. However, because the salient-pole rotor is a compound excitation rotor, which is simply equivalent to the surface-mount PM rotor, the accuracy is poor, so this paper proposes a segmented equivalent method, equating the electrical excitation and V-shaped PM segments as surface-mount PMs, and simulating the synthesis of the electric excitation magnetic field and the V-shaped PM magnetic field by determining the distribution scale coefficient. The improved subdomain method can accurately reflect the distribution of magnetic flux density in each subdomain, and can further improve the solution accuracy by increasing the number of segments, and further analyze other characteristics of the compound excitation pole generator based on the magnetic flux density analysis model of each subdomain.

Reasonable assumptions can simplify the model and reflect the essence of the generator model. If too many secondary factors are considered, it will make it more difficult to establish the generator model. To simplify the mathematical model, the following assumptions can be made [21]:

- Ignore the vortex effect and end effect;
- The demagnetization curve of the PM material is linear;
- The current density of the coil edge in the stator slot is evenly distributed and has only the component in the *z*-axis direction;
- The stator rotor core magnetic permeability is infinite.

At the same time, due to the small size of the stator slot, for the convenience of calculation, the stator slot shoulder and the arc part at the bottom of the stator slot are negligible.

For analytical calculations of subdomain models, having standard boundary conditions within a two-dimensional coordinate plane is essential. To match the boundary conditions of Poisson's equation, the built-in V-shaped PM is equivalent to the electrical excitation winding as a sector-shaped region, but due to the variability of the electro-excitation magnetic field, the excitation source is again divided into two regions, so that the outer boundary of the sectoral region is a segmented arc.

$$\begin{cases} \frac{B_r a_{r2}}{\mu_0 \mu_1} + \frac{B_r h_{r2}}{\mu_0 \mu_1} + k_{dx} N_r I = \frac{B_r h_{m3}}{\mu_0 \mu_1}, \frac{\pi - 6\theta_0}{12} < \alpha \le \frac{\pi + 6\theta_0}{12} \\ (1 - k_{dx}) N_r I = \frac{B_r h_{m3}}{\mu_0 \mu_1}, 0 < \alpha \le \frac{\pi - 6\theta_0}{12} \cup \frac{\pi + 6\theta_0}{12} < \alpha \le \frac{\pi}{6} \end{cases} \tag{17}$$

where $\mu_1$ is the relative permeability of Nd-Fe-B, $a_{r2}$ is the thickness of the magnetization direction of the V-shaped PM, $h_{r2}$ is the length of the magnetic flux path of the PM magnetic field in the rotor, $h_{m3}$ is the equivalent length of the sector PM in the direction of magnetization, $\theta_0$ is the mechanical angle occupied by the V-shaped PM, $N_r$ is the number of turns for

the excitation windings, $k_{dx}$ is the coefficient of electrical excitation distribution, $k_{dx} = 0.11$, $I$ is the excitation current.

The salient-pole generator is divided into four subdomains: stator slot (subdomain I), stator slot notch (subdomain II), air gap (subdomain III), and PM (subdomain IV). A schematic diagram of the salient-pole generator area division is shown in Figure 8.

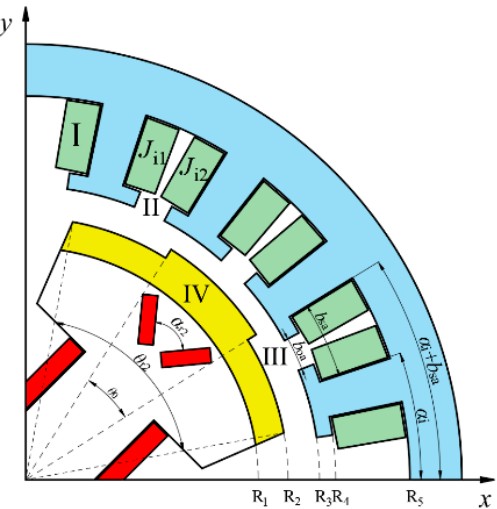

**Figure 8.** Schematic diagram of the regional division of the salient-pole generator.

Where $R_1$–$R_5$ is the radius of the corresponding boundary, $b_{oa}$ is the mechanical angle occupied by the equivalent posterior stator notch, $b_{sa}$ is the equivalent mechanical angle occupied inside the stator slot. Depending on the excitation source, the magnetic field control equations of each subdomain are different, and the vector magnetic quantity of the stator slot (subdomain I) and the PM (subdomain IV) satisfies the Poisson's equation, and the stator slot notch (subdomain II) and the air gap (subdomain III) satisfy the Laplace's equation [22].

Through the Poisson's equation and Laplace's equation, the expression of the partial differential equation for each subdomain can be obtained, and then the general solution of each subdomain can be obtained by separating the variable method and Fourier decomposition, and finally the boundary condition can be brought in to find the unique solution of each subdomain, and the solution process is shown as follows.

### 2.2.1. Stator Slot (Subdomain I)

The vector magnetic quantity of the stator slot satisfies Poisson's equation, and its partial differential equation is expressed as follows:

$$\begin{cases} \frac{B_r a_{r2}}{\mu_0 \mu_1} + \frac{B_r h_{r2}}{\mu_0 \mu_1} + k_{dx} N_r I = \frac{B_r h_{m3}}{\mu_0 \mu_1}, \frac{\pi - 6\theta_0}{12} < \alpha \leq \frac{\pi + 6\theta_0}{12} \\ (1 - k_{dx}) N_r I = \frac{B_r h_{m3}}{\mu_0 \mu_1}, 0 < \alpha \leq \frac{\pi - 6\theta_0}{12} \cup \frac{\pi + 6\theta_0}{12} < \alpha \leq \frac{\pi}{6} \end{cases} \quad (18)$$

where the boundary conditions of Poisson's equation are as follows:

$$\begin{cases} \frac{\partial A_{zIi}}{\partial r}\Big|_{r=R_4} = 0, \ \frac{\partial A_{zIi}}{\partial \alpha}\Big|_{\alpha=\alpha_i} = 0, \ \frac{\partial A_{zIi}}{\partial \alpha}\Big|_{\alpha=\alpha_i+b_{sa}} = 0 \\ \frac{\partial A_{zIi}}{\partial r}\Big|_{r=R_4} = \frac{\partial A_{zIIi}}{\partial r}\Big|_{r=R_4} \end{cases}$$

Since the magnetic permeability of the stator core is assumed to be infinite, the current density in the stator slot is mirrored by the left and right boundaries of the slot and becomes a periodic signal of a periodicity. The stator slot current density distribution is shown in Figure 9.

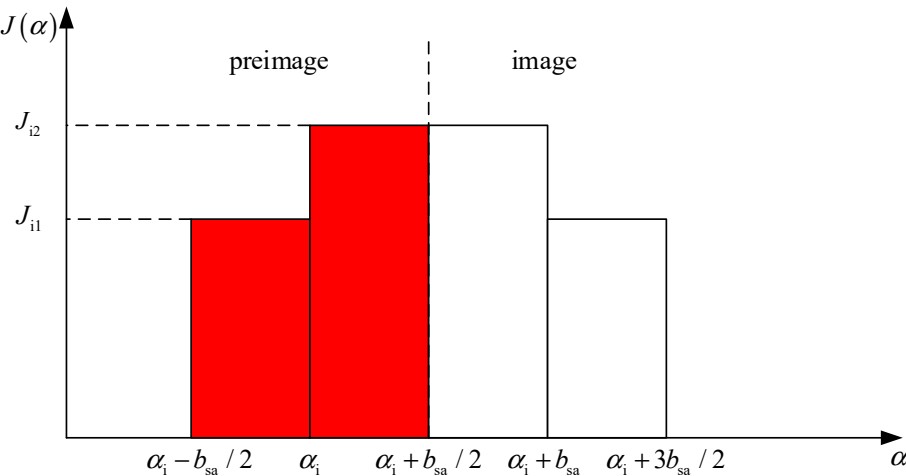

**Figure 9.** Stator slot current density distribution.

Figure 9 shows the distribution of the current density in the stator over a period after the mirror image, which is expanded by the Fourier series [23]:

$$J = \frac{J_{i1} + J_{i2}}{2} + \sum_{n} \frac{2}{n\pi} (J_{i1} + J_{i2}) \sin(n\pi/2) \cos\left[\frac{n\pi}{b_{sa}}(\alpha + b_{sa} - \alpha_i)\right] \qquad (19)$$

where $J_{i1}$ and $J_{i2}$ are the current density of the two sets of windings in the slot, $n$ is the logarithm of spatial harmonics within the stator slot subdomain, and $\alpha$ is circumferential angle.

Using the separation variable method, the general solution of the stator slot subdomain can be solved as shown below [24]:

$$A_{zIi} = -\frac{\mu_0(J_{i1} + J_{i2})}{8}r^2 \ln r + \sum_{n} \left\{ \left[ A^I \left(\frac{R_4}{R_5}\right)\left(\frac{r}{R_5}\right)^n + B^I \left(\frac{r}{R_4}\right)^n \right] \cos\left[\left(\frac{n\pi}{b_{sa}}\right)(\alpha + b_{sa} - \alpha_i)\right] + \frac{2\mu_0(J_{i1}+J_{i2})\sin(n\pi/2)}{n\pi\left[\left(\frac{n\pi}{b_{sa}}\right)^2 - 4\right]} \left[r^2 - \frac{2}{E_n}R_5^2\left(\frac{r}{R_5}\right)^n\right] \cos\left[\left(\frac{n\pi}{b_{sa}}\right)(\alpha + b_{sa} - \alpha_i)\right] \right\} \qquad (20)$$

where $A^I$ and $B^I$ are the harmonic coefficients of the stator slot subdomain, $R_4$ is the circumferential radius of the stator slot bottom, and $R_5$ is the circumferential radius of the stator slot top.

### 2.2.2. Stator Slot Notch (Subdomain III)

The vector magnetic quantity of the stator slot notch satisfies Laplace's equation, and its partial differential equation is expressed as:

$$\begin{cases} \frac{\partial^2 A_{zIIi}}{\partial r^2} + \frac{1}{r}\frac{\partial A_{zIIi}}{\partial r} + \frac{1}{r^2}\frac{\partial^2 A_{zIIi}}{\partial \alpha^2} = 0 \\ R_3 < r < R_4, \alpha_i + \frac{b_{sa}-b_{oa}}{2} < \alpha < \alpha_i + \frac{b_{sa}+b_{oa}}{2} \end{cases} \qquad (21)$$

where the boundary conditions of Laplace's equation are as follows:

$$\begin{cases} \left.\frac{\partial A_{zIIi}}{\partial \alpha}\right|_{\alpha=\alpha_i + \frac{b_{sa}-b_{oa}}{2}} = 0, \ \left.\frac{\partial A_{zIIi}}{\partial \alpha}\right|_{\alpha=\alpha_i + \frac{b_{sa}+b_{oa}}{2}} = 0 \\ \left.\frac{\partial A_{zIIi}}{\partial r}\right|_{r=R_3} = \left.\frac{\partial A_{zIIIi}}{\partial r}\right|_{r=R_3}, \ \left.\frac{\partial A_{zIIi}}{\partial r}\right|_{r=R_4} = \left.\frac{\partial A_{zIi}}{\partial r}\right|_{r=R_4} \end{cases}$$

Using the separation of variables method, the stator slot notch subdomain can be solved as follows:

$$A_{zIIi} = \sum_{m} \left\{ \left[ A^{II}\left(\frac{r}{R_4}\right)^m + B^{II}\left(\frac{r}{R_3}\right)^{-m} \right] \cos\left[m\left(\alpha + \frac{b_{oa}}{2} - \alpha_i\right)\right] + \left[ C^{II}\left(\frac{r}{R_4}\right)^m - D^{II}\left(\frac{r}{R_3}\right)^{-m} \right] \sin\left[m\left(\alpha + \frac{b_{oa}}{2} - \alpha_i\right)\right] \right\} \qquad (22)$$

where $A^{II}$, $B^{II}$, $C^{II}$, and $D^{II}$ are the harmonic coefficient of the stator slot notch subdomain, $m$ is the number of spatial harmonics in the stator slot notch subdomain, $R_3$ is the circumferential radius of the stator slot notch.

### 2.2.3. Air Gap (Subdomain III)

The vector magnetic quantity of the air gap satisfies Laplace's equation, and its partial differential equation is expressed as:

$$
\begin{cases}
\frac{\partial^2 A_{\mathrm{zIIIi}}}{\partial r^2} + \frac{1}{r}\frac{\partial A_{\mathrm{zIIIi}}}{\partial r} + \frac{1}{r^2}\frac{\partial^2 A_{\mathrm{zIIIi}}}{\partial \alpha^2} = 0 \\
R_2 < r < R_3, 0 < \alpha < \frac{\pi}{6}
\end{cases}
\tag{23}
$$

where the boundary conditions of Laplace's equation are as follows:

$$
\begin{cases}
\left.\frac{\partial A_{\mathrm{zIIIi}}}{\partial r}\right|_{r=R_2} = \left.\frac{\partial A_{\mathrm{zIVi}}}{\partial \alpha}\right|_{r=R_2} \\
\left.\frac{\partial A_{\mathrm{zIIIi}}}{\partial r}\right|_{r=R_3} = \begin{cases}
\left.\frac{\partial A_{\mathrm{zIIi}}}{\partial \alpha}\right|_{r=R_3}, \alpha_i + \frac{b_{\mathrm{sa}}-b_{\mathrm{oa}}}{2} < \alpha < \alpha_i + \frac{b_{\mathrm{sa}}+b_{\mathrm{oa}}}{2} \\
\left(1 - \frac{\mu_1 \delta}{\mu_0 h_{\mathrm{m3}}}\right)\left.\frac{\partial A_{\mathrm{zIVi}}}{\partial \alpha}\right|_{r=R_2}, \text{other}
\end{cases}
\end{cases}
$$

Using the separation of variables method, the air gap subdomain can be solved as follows:

$$
A_{\mathrm{zIIIi}} = \sum_k \left[ A^{\mathrm{III}}\left(\frac{r}{R_4}\right)^k + B^{\mathrm{III}}\left(\frac{r}{R_3}\right)^{-k}\right]\cos(k\alpha) + \sum_k\left[ C^{\mathrm{III}}\left(\frac{r}{R_4}\right)^k + D^{\mathrm{III}}\left(\frac{r}{R_3}\right)^{-k}\right]\sin(k\alpha)
\tag{24}
$$

where $A^{\mathrm{III}}$, $B^{\mathrm{III}}$, $C^{\mathrm{III}}$, and $D^{\mathrm{III}}$ are the harmonic coefficients for the air gap subdomain, $k$ is the logarithm of spatial harmonics in the air gap subdomain, and $R_3$ is the inner radius of the stator.

### 2.2.4. Permanent Magnet (Subdomain IV)

The vector magnetic quantity of the PM satisfies Poisson's equation, and its partial differential equation is expressed as:

$$
\begin{cases}
\frac{\partial^2 A_{\mathrm{zIVi}}}{\partial r^2} + \frac{1}{r}\frac{\partial A_{\mathrm{zIVi}}}{\partial r} + \frac{1}{r^2}\frac{\partial^2 A_{\mathrm{zIVi}}}{\partial \alpha^2} = -\frac{\mu_0}{r}\left(M_\alpha - \frac{\partial M_r}{\partial \alpha}\right) \\
R_1 < r < R_2, \frac{\pi - 2\theta_{\mathrm{r2}}}{12} < \alpha < \frac{\pi + 10\theta_{\mathrm{r2}}}{12}
\end{cases}
\tag{25}
$$

where the boundary conditions of Poisson's equation are as follows:

$$
\left.\frac{\partial A_{\mathrm{zIVi}}}{\partial \alpha}\right|_{\alpha=\frac{\pi-2\theta_{\mathrm{r2}}}{12}} = 0 , \quad \left.\frac{\partial A_{\mathrm{zIVi}}}{\partial \alpha}\right|_{\alpha=\frac{\pi+10\theta_{\mathrm{r2}}}{12}} = 0 , \quad \left.\frac{\partial A_{\mathrm{zIVi}}}{\partial r}\right|_{r=R_1} = \left.\frac{\partial A_{\mathrm{zIIIi}}}{\partial r}\right|_{r=R_1}
$$

When the permanent magnet is magnetized, Fourier decomposes under a pair of magnetic poles to obtain the radial and tangential components of the magnetization intensity of the permanent magnet as follows [25]:

$$
\begin{cases}
M_{\mathrm{r}} = \sum\limits_{n=1,3,5} M_{\mathrm{rl}}\cos\left[ l\left(\alpha + \frac{\theta_0 R_1}{2R_2} - \alpha_i\right)\right] \\
M_{\alpha} = \sum\limits_{n=1,3,5} M_{\alpha l}\sin\left[ l\left(\alpha + \frac{\theta_0 R_1}{2R_2} - \alpha_i\right)\right]
\end{cases}
\tag{26}
$$

where $M_{\mathrm{rl}}$ and $M_{\alpha l}$ are the harmonic components of the tangential and radial components of the residual magnetization intensity of the PM, $l$ is the logarithm of spatial harmonics in the PM subdomain, $\theta_{\mathrm{r2}}$ is the mechanical angle occupied by each pole of the salient-pole rotor.

For parallel magnetization, the following equation can be used:

$$
\begin{cases}
M_{\mathrm{rl}} = \frac{2pB_{\mathrm{r}}\sin\frac{(l+1)k_{\mathrm{r2}}}{2p}}{(l+1)\mu_0} \\
M_{\alpha l} = \frac{2pB_{\mathrm{r}}\sin\frac{(l-1)k_{\mathrm{r2}}}{2p}}{(l-1)\mu_0}
\end{cases}
\tag{27}
$$

where $k_{r2}$ is the arc coefficient of the salient-pole rotor and $B_r$ is the remanence of the PM. Using the separation of variables method, the PM subdomain can be solved as follows:

$$A_{zIVi} = A^{IV} + B^{IV} \ln r + \sum_l \left[ C^{IV} \left( \frac{r}{R_3} \right)^l + D^{IV} \left( \frac{r}{R_2} \right)^{-l} \right] \cos \left( \alpha + \frac{\theta_0 R_1}{2R_3} - \alpha_i \right) + \sum_l \mu_0 \left( \frac{M_{rl} - M_{\alpha l}}{2} \ln r \right) \cos \left( \frac{l\pi}{b_{oa}} \right) \left( \alpha + \frac{\theta_0 R_1}{2R_3} - \alpha_i \right) \quad (28)$$

where $A^{IV}$, $B^{IV}$, $C^{IV}$, and $D^{IV}$ are harmonic coefficients for the PM subdomain.

### 2.2.5. Pending Coefficient

According to the basic principle that the normal magnetic density and tangential magnetic field strength at the interface of adjacent subdomains are equal, the harmonic coefficients in the magnetic vector of each subdomain can be solved, and the expression is shown in Equation (29).

$$\begin{cases} A_{zIi}|_{r=R_4} - A_{zIIi}|_{r=R_4} = 0, \alpha_i + \frac{b_{sa} - b_{oa}}{2} < \alpha < \alpha_i + \frac{b_{sa} + b_{oa}}{2} \\ H_{zI}|_{r=R_4} - H_{zII}|_{r=R_4} = 0, \alpha_i + \frac{b_{sa} - b_{oa}}{2} < \alpha < \alpha_i + \frac{b_{sa} + b_{oa}}{2} \\ A_{zIIi}|_{r=R_3} - A_{zIIIi}|_{r=R_3} = 0, \alpha_i + \frac{b_{sa} - b_{oa}}{2} < \alpha < \alpha_i + \frac{b_{sa} + b_{oa}}{2} \\ H_{zIIi}|_{r=R_3} - H_{zIIIi}|_{r=R_3} = 0, \alpha_i + \frac{b_{sa} - b_{oa}}{2} < \alpha < \alpha_i + \frac{b_{sa} + b_{oa}}{2} \\ A_{zIIIi}|_{r=R_2} - A_{zIVi}|_{r=R_2} = 0, \frac{\pi - 2\theta_{r2}}{12} < \alpha < \frac{\pi + 10\theta_{r2}}{12} \\ H_{zIIIi}|_{r=R_2} - H_{zIVi}|_{r=R_2} = 0, \frac{\pi - 2\theta_{r2}}{12} < \alpha < \frac{\pi + 10\theta_{r2}}{12} \end{cases} \quad (29)$$

### 2.2.6. Analytical Model of the Magnetic Field in Each Subdomain

Depending on the relationship between magnetic flux density and magnetic vector position, the radial and tangential components of magnetic flux density can be expressed as:

$$\begin{aligned} B_r &= \frac{1}{r} \frac{\partial A}{\partial \alpha} \\ B_\alpha &= -\frac{\partial A}{\partial r} \end{aligned} \quad (30)$$

According to the magnetic vector pass solution of each subdomain derived above, the magnetic flux density expression of each subdomain can be obtained.

The radial and tangential components of the magnetic flux density in the stator slot subdomain are shown below:

$$B_{zIr} = -E_n \sum_n \left\{ \left[ A^I \left( \frac{R_4}{R_5} \right) \left( \frac{r}{R_5} \right)^n + B^I \left( \frac{r}{R_4} \right)^n \right] \sin \left[ \left( \frac{n\pi}{b_{sa}} \right) (\alpha + b_{sa} - \alpha_i) \right] - \mu_0 \frac{n\pi}{b_{sa}r} \frac{2(J_{i1} + J_{i2}) \sin(n\pi/2)}{n\pi \left[ \left( \frac{n\pi}{b_{sa}} \right)^2 - 4 \right]} \left[ r^2 - \frac{2}{E_n} R_5^2 \left( \frac{r}{R_4} \right)^n \right] \sin \left[ \left( \frac{n\pi}{b_{sa}} \right) (\alpha + b_{sa} - \alpha_i) \right] \right\} \quad (31)$$

$$\begin{aligned} B_{zI\alpha} &= \frac{\mu_0}{2} J_{i0} r \ln r + \frac{\mu_0}{4} J_{i0} r \\ &- \sum_n \left\{ n \left[ A^I \left( \frac{R_4}{R_5} \right) \left( \frac{r}{R_5} \right)^{n-1} + B^I \left( \frac{r}{R_4} \right)^{n-1} \right] \cos[E_n(\alpha + b_{sa} - \alpha_i)] - \mu_0 \frac{2(J_{i1} + J_{i2}) \sin(n\pi/2)}{n\pi \left[ \left( \frac{n\pi}{b_{sa}} \right)^2 - 4 \right]} \left[ 2r - \frac{2}{E_n} n R_5^2 \left( \frac{r}{R_4} \right)^{n-1} \right] \cos \left[ \left( \frac{n\pi}{b_{sa}} \right) (\alpha + b_{sa} - \alpha_i) \right] \right\} \end{aligned} \quad (32)$$

The radial and tangential components of the magnetic flux density in the stator slot notch subdomain are shown below:

$$B_{zIIr} = -\frac{m}{r} \sum_m \left\{ \left[ A^{II} \left( \frac{r}{R_4} \right)^m + B^{II} \left( \frac{r}{R_3} \right)^{-m} \right] \sin[m(\alpha + b_{oa} - \alpha_i)] - \frac{m}{r} \left[ C^{II} \left( \frac{r}{R_4} \right)^m - D^{II} \left( \frac{r}{R_3} \right)^{-m} \right] \cos[m(\alpha + b_{oa} - \alpha_i)] \right\} \quad (33)$$

$$B_{zII\alpha} = -\sum_m \left\{ m \left[ A^{II} \left( \frac{r}{R_4} \right)^{m-1} - B^{II} \left( \frac{r}{R_3} \right)^{-m-1} \right] \cos[m(\alpha + b_{oa} - \alpha_i)] - m \left[ C^{II} \left( \frac{r}{R_4} \right)^{m-1} + D^{II} \left( \frac{r}{R_3} \right)^{-m-1} \right] \sin[m(\alpha + b_{oa} - \alpha_i)] \right\} \quad (34)$$

The radial and tangential components of the magnetic flux density in the air gap subdomain are shown below:

$$B_{zIIIr} = -\frac{k}{r} \sum_k \left[ A^{III} \left( \frac{r}{R_3} \right)^k + B^{III} \left( \frac{r}{R_2} \right)^{-k} \right] \sin(k\alpha) + \frac{k}{r} \sum_k \left[ C^{III} \left( \frac{r}{R_3} \right)^k + D^{III} \left( \frac{r}{R_2} \right)^{-k} \right] \cos(k\alpha) \quad (35)$$

$$B_{zIII\alpha} = -\sum_k \left[ kA^{III}\left(\frac{r}{R_3}\right)^{k-1} - kB^{III}\left(\frac{r}{R_2}\right)^{-k-1} \right]\cos(k\alpha) - \sum_k \left[ kC^{III}\left(\frac{r}{R_3}\right)^{k-1} - kD^{III}\left(\frac{r}{R_2}\right)^{-k-1} \right]\sin(k\alpha) \quad (36)$$

The radial and tangential components of the magnetic flux density in the PM subdomain are shown below:

$$B_{zIVr} = -\frac{1}{r}\sum_l \left[ C^{IV}\left(\frac{r}{R_2}\right)^l + D^{IV}\left(\frac{r}{R_1}\right)^{-l} \right]\sin\left(\alpha + \frac{\theta_0 R_1}{2R_2} - \alpha_i\right) - \frac{1}{r}\frac{l\pi}{b_{oa}}\sum_l \mu_0\left(\frac{M_{rl} - M_{\alpha l}}{2}\ln r\right)\sin\left(\frac{l\pi}{b_{oa}}\right)\left(\alpha + \frac{\theta_0 R_1}{2R_2} - \alpha_i\right) \quad (37)$$

$$B_{zIV\alpha} = -\frac{B^{IV}}{r} - \sum_l \left[ lC^{IV}\left(\frac{r}{R_2}\right)^{l-1} - lD^{IV}\left(\frac{r}{R_1}\right)^{-l-1} \right]\cos\left(\alpha + \frac{\theta_0 R_1}{2R_2} - \alpha_i\right) - \sum_l \mu_0\left(\frac{M_{rl} - M_{\alpha l}}{2r}\right)\cos\left(\frac{l\pi}{b_{oa}}\right)\left(\alpha + \frac{\theta_0 R_1}{2R_2} - \alpha_i\right) \quad (38)$$

*2.3. Loss Analysis of Hybrid Excitation Generator*

Based on the analysis of the magnetic field of the hybrid magnetic circuit generator, the mechanism of generator loss is analyzed. High efficiency is one of the important design indicators of the generator. The loss of the hybrid magnetic circuit generator is an important factor affecting its efficiency. The generator will produce different loss during operation. The loss mainly includes core loss, winding copper loss, permanent magnet eddy current loss and mechanical loss, among which copper loss and iron loss account for a large proportion.

The expression of core loss is as follows [26]:

$$P_{Fe} = \frac{d\sigma\pi^2}{2\sqrt{\mu\sigma\pi}}\sum_{n=1}^{\infty} f^2(B_{Fer}^2 + B_{Fet}^2)\gamma(f) + k_e\sum_{n=1}^{\infty} nf(B_{Fer}^2 + B_{Fet}^2) + \frac{k_x}{T}\int_0^T (B_{Fer}^2 + B_{Fet}^2)^{3/4}dt \quad (39)$$

where $\mu$ is the permeability, $f$ is the fundamental frequency, $\gamma(f)$ is the coefficient of skin effect, $B_{Fer}$ and $B_{Fet}$ are the radial and tangential component of magnetic flux density, $k_e$ is the eddy current loss correction factor, and $k_x$ is the additional loss correction factor.

Copper loss can be expressed according to the empirical formula as follows [27]:

$$P_c = 3 \times I_m^2 R = 3 \times I_m^2 \frac{\rho_c l_o}{S_c} \times \frac{N_z}{N_c} a N_m \quad (40)$$

where $I_m$ is the effective value of current, $R$ is the phase resistance, $\rho_c$ is the resistivity of copper, $l_o$ is the length of each coil, $S_c$ is the cross-sectional area of a single copper conductor, $N_z$ is the number of coil turns, $N_c$ is the number of parallel windings of the coil, $a$ is the number of parallel circuits of each phase winding, and $N_m$ is the number of coils in series.

## 3. Performance Optimization of the Dual-Rotor Hybrid Excitation Generator

Similarly, the performance optimization of the generator can be considered separately from two aspects. According to the different characteristics of the two rotors, different methods are used to optimize the structure of the two rotors.

*3.1. Multi-Objective Optimization of the Claw Pole Rotor*

In this paper, the multi-objective optimization design is carried out by using the sample point acquisition-Pareto frontier solution method for the claw pole rotor. Among them, the sample point acquisition algorithm mainly includes the full factor method, orthogonal experimental method, Latin hypercube method, and other methods [28]. For a large number of design variables, the full factor method and orthogonal experimental method involve too many sampling points, which will greatly reduce the optimization efficiency, so the Latin hypercube method can cover the entire sampling space with a small number of sample points is selected, and the currently optimized Latin hypercube sampling method can make the sampling point distribution more uniform [29].

3.1.1. Design Variables and Optimization Goals

According to the different sensitivity of each parameter to the generator induction EMF, the parameters of the claw pole tooth tip arc coefficient, the claw pole tooth root arc

coefficient, the claw pole tooth tip thickness, the claw pole tooth root thickness, the flange thickness, and the radial thickness of the inter-pole PM are selected as the design variables, and the fundamental amplitude of the generator induction EMF and the sine waveform distortion rate are two optimization targets. The constraints for each design variable are shown in Table 1.

**Table 1.** Optimization constraints for each design variable.

| Design Variables | Initial Value | Constraints |
|---|---|---|
| The arc coefficient of the claw tooth cusp pole $k_{cj}$ | 0.3 | $0.2 \leq k_{cj} \leq 0.5$ |
| The arc coefficient of the root of the claw tooth $k_{cg}$ | 1.1 | $0.9 \leq k_{cg} \leq 1.2$ |
| Claw pole tooth tip thickness $h_{cj}$/mm | 2.5 | $2 \leq h_{cj} \leq 3.5$ |
| Claw tooth root thickness $h_{cg}$/mm | 6 | $5 \leq h_{cg} \leq 7$ |
| Flange thickness $h_{flp}$/mm | 5 | $5 \leq h_{flp} \leq 6.5$ |
| The radial thickness of the inter-pole PM $h_{jyc}$/mm | 4 | $3 \leq k_{jyc} \leq 5$ |

The optimization goal is to maximize the induced EMF fundamental wave and minimize the distortion rate of the sine waveform under the condition of satisfying the constraints, and the optimization model is shown as follows:

$$\begin{cases} \max U_x(x_1, x_2, \ldots, x_m) \\ \min THD_u(x_1, x_2, \ldots, x_m) \end{cases} \tag{41}$$

where $U_x$ is the amplitude of the induced EMF fundamental wave and $THD_u$ is the induced EMF waveform distortion rate. The two arguments are functions of 6 design variables.

3.1.2. Fundamentals of Latin Hypercube Sampling

Assume that there are $M$ random variables in an optimization problem and $X_m$ is any of them. The distribution function is:

$$Y_m = F_m(X_m) \tag{42}$$

Assuming the sampling number is $N$, the longitudinal axis of $Y_m$ is divided into $N$ equal intervals. It is assumed that each variable is independent and $x_{mn}$ is the $n$th sampling value of the $m$th variable. Then, the Latin hypercube sampling procedure is as follows [30,31]:

- Generate an M × N dimension matrix $L_{M\times N}$. Each row of the matrix is a random sequence of (1, $N$) integers, and $a_{mn}$ is its element in a row $m$ and column $n$.
- Generate an M × N dimension matrix $U_{M\times N}$. Each element of the matrix is evenly distributed [0, 1], and $u_{mn}$ is its element in a row $m$ and column $n$.
- The M × N dimensional sampling matrix $X_{M\times N}$ is calculated, and $x_{mn}$ is its element in a row $m$ and column $n$.

$$x_{mn} = F_m^{-1}\left(\frac{a_{mn} - u_{mn}}{N}\right) \tag{43}$$

where $m = 1, 2, \ldots, M$; $n = 1, 2, \ldots, N$. The schematic diagram of the Latin hypercube sampling is shown in Figure 10.

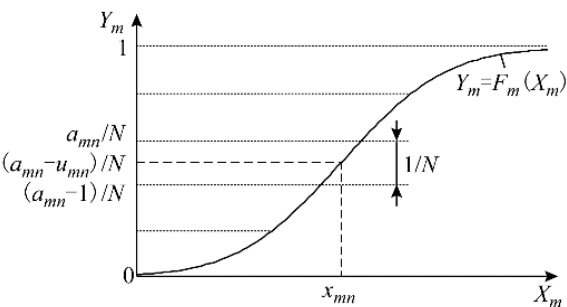

**Figure 10.** Schematic diagram of Latin hypercube sampling.

Due to the particularity of the claw pole rotor structure, a three-dimensional model needs to be established when performing finite element simulation calculations, which requires a large amount of calculation. Therefore, the Latin superposition method of optimizing the number of sample points is 100, and the samples are collected from the design variables.

### 3.1.3. The Fitting Relationship between Each Design Variable and Response Value

Simulation analysis of 100 sample points yields a diagram of the fitted surface between the design variables and the response values. Some of the design variables fit the response value diagram as shown in Figure 11.

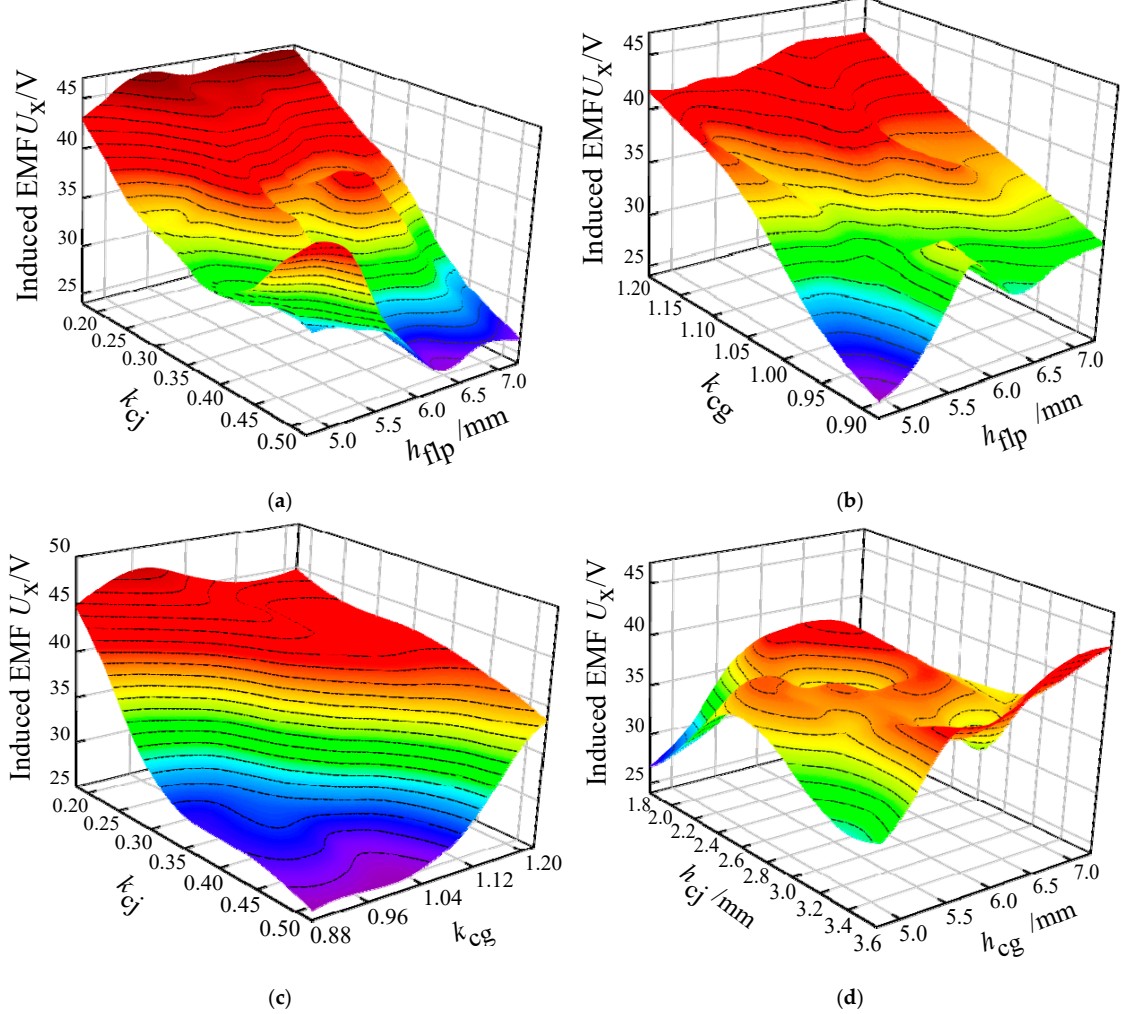

**Figure 11.** *Cont.*

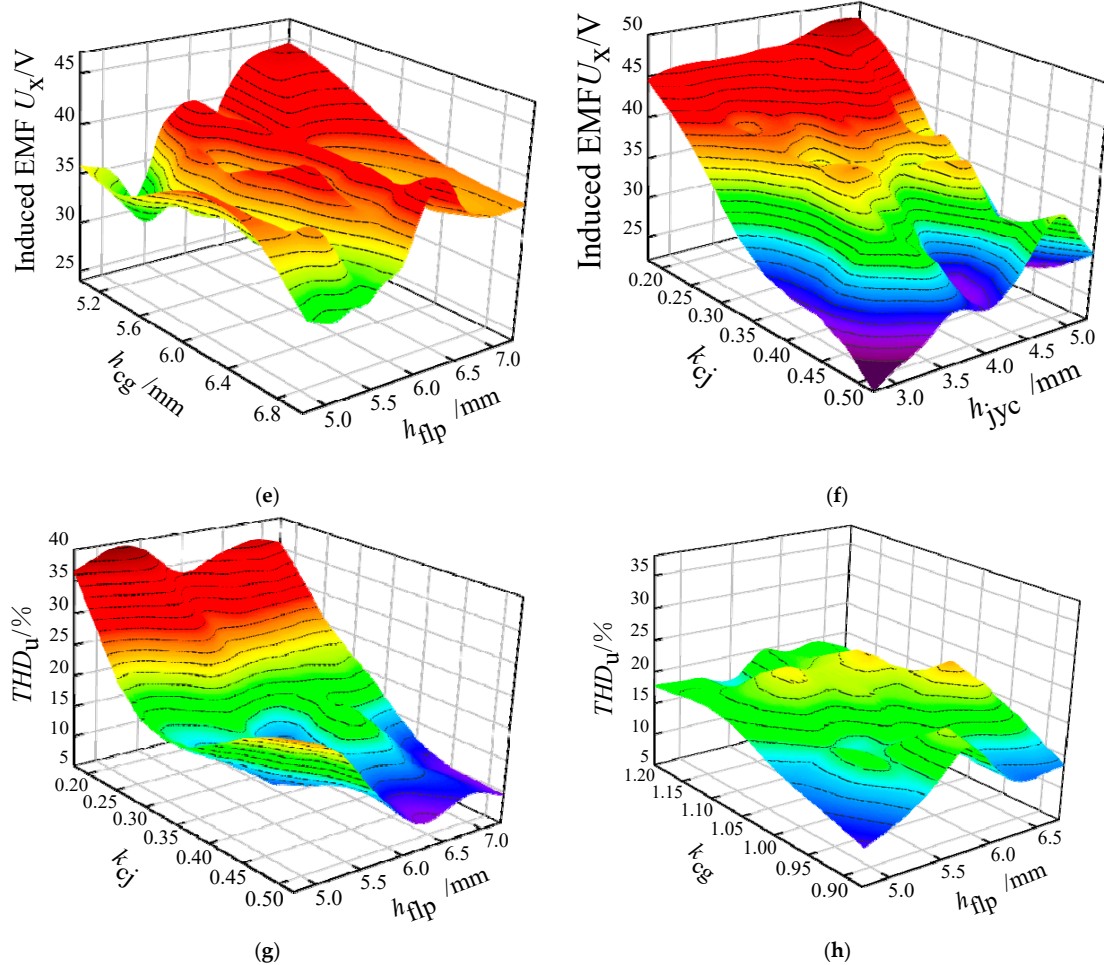

**Figure 11.** Design variable and response value fitting diagram (partial). (**a**) $k_{cj}$, $h_{flp}$, and $U_x$; (**b**) $k_{cg}$, $h_{flp}$, and $U_x$; (**c**) $k_{cj}$, $k_{cg}$, and $U_x$; (**d**) $h_{cj}$, $h_{cg}$, and $U_x$; (**e**) $h_{cg}$, $h_{flp}$, and $U_x$; (**f**) $k_{cj}$, $h_{jyc}$, and $U_x$; (**g**) $k_{cj}$, $h_{flg}$, and $THD_u$; (**h**) $k_{cg}$, $h_{flp}$, and $THD_u$.

As can be seen from Figure 11, in addition to the approximate unmodalaline and linear relationship between $k_{cg}$-$k_{cj}$, $k_{cg}$-$h_{flp}$, $k_{cg}$-$h_{flp}$ and the response value, the spatial relationship between other design variables and the response value has the characteristics of multi-peak, multi-valley, non-uniform, and nonlinear, which further reveals the possibility that the multi-objective optimization model has multiple optimal solutions. The fitting relationship between the design variable and the response value can obtain the influence characteristics of each design variable on the response value, which can be represented by the sensitivity of each design variable to the change in the response value. The sensitivity of each design variable is shown in Table 2, the histogram of the sensitivity of each design variable to $U_x$ and $THD_u$ is shown in Figure 12.

**Table 2.** Sensitivity of design variables concerning response values.

| Design Variables | Sensitivity for $U_x$ | Sensitivity for $THD_u$ |
|:---:|:---:|:---:|
| $k_{cj}$ | 0.55276 | 0.56532 |
| $k_{cg}$ | 0.38321 | 0.15872 |
| $h_{flp}$ | 0.09483 | 0.02394 |
| $h_{cj}$ | 0.05397 | 0.07471 |
| $h_{cg}$ | 0.05216 | 0.03763 |
| $h_{jyc}$ | 0.02274 | 0.08146 |

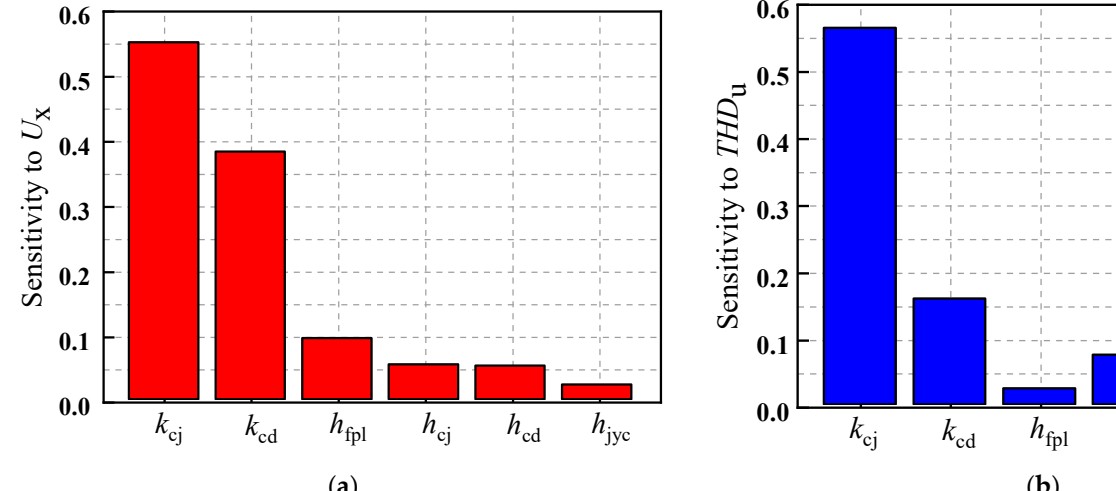

**Figure 12.** The sensitivity of optimization objective. (**a**) Sensitivity of each design variable to $U_x$; (**b**) sensitivity of each design variable to $THD_u$.

As can be seen from Table 2 and Figure 12, the design variables $k_{cj}$, $k_{cg}$, and $h_{flp}$ have a higher sensitivity to the response value $U_x$, and the design variables $k_{cj}$ and $k_{cg}$ have a higher sensitivity to the response value $THD_u$. Overall, the design variables $k_{cj}$ and $k_{cg}$ have the highest sensitivity to the two response values. For the claw pole rotor optimization model, the six design variables $k_{cg}$, $k_{cj}$, $h_{cj}$, $h_{cd}$, $h_{flp}$, and $h_{jyc}$ can be reduced to two design variables $k_{cj}$ and $k_{cg}$. The dimensionality reduction of the optimization model can greatly improve the calculation speed of the model and determine the optimal parameters more flexible.

### 3.1.4. Pareto Frontier Solution

The Pareto optimization method is a common method for solving multi-objective optimization problems, especially for dual-objective optimal problems, and the Pareto frontier obtained by solving can intuitively reflect the distribution law of the optimal solution set from two-dimensional space [32]. The traditional Pareto frontier is constrained by multiple optimization target minimums [33]. However, in the optimization model, the induced EMF fundamental amplitude should be taken at its maximum, so the induced EMF is negative in the actual solution. The Pareto frontier distribution is shown in Figure 13.

As can be seen from Figure 13, there are 12 sample points on the Pareto frontier, which are valid solutions for the optimization model, and the specific values of the design variables and optimization targets are shown in Table 3.

**Table 3.** The concrete values of design variables and optimization goals in valid solutions.

| Serial Number | $k_{cg}$ | $k_{cj}$ | $h_{cj}$/mm | $h_{cg}$/mm | $h_{flp}$/mm | $h_{jyc}$/mm | $THD_u$/% | $U_x$/V |
|---|---|---|---|---|---|---|---|---|
| 48 | 1.0245 | 0.3815 | 2.0375 | 6.0875 | 5.29 | 4.73 | 6.66 | 34.747 |
| 74 | 0.9885 | 0.3485 | 2.0225 | 5.5025 | 6.69 | 3.31 | 8.96 | 35.488 |
| 24 | 1.0485 | 0.4805 | 3.0875 | 6.4025 | 6.79 | 4.51 | 9.54 | 36.059 |
| 37 | 1.0305 | 0.3935 | 3.4325 | 6.4625 | 5.93 | 3.03 | 10.52 | 36.534 |
| 8 | 0.9615 | 0.3665 | 2.9075 | 5.9675 | 6.05 | 4.55 | 11.44 | 36.608 |
| 34 | 1.1745 | 0.3245 | 3.1025 | 5.3375 | 6.03 | 4.31 | 12.36 | 40.793 |
| 68 | 1.1445 | 0.3395 | 2.0975 | 6.3125 | 5.85 | 4.91 | 14.16 | 42.482 |
| 18 | 1.0335 | 0.3365 | 2.6375 | 5.7425 | 6.61 | 4.41 | 18.84 | 42.620 |
| 35 | 0.9945 | 0.2645 | 3.4775 | 6.2225 | 6.53 | 4.29 | 25.28 | 42.685 |
| 38 | 1.0215 | 0.2825 | 2.9975 | 5.3675 | 6.75 | 4.75 | 26.78 | 42.887 |
| 90 | 1.0815 | 0.2585 | 2.9675 | 5.6675 | 6.31 | 4.99 | 27.38 | 43.796 |
| 98 | 0.9315 | 0.2405 | 3.4175 | 5.5625 | 6.17 | 4.57 | 33.3 | 44.960 |

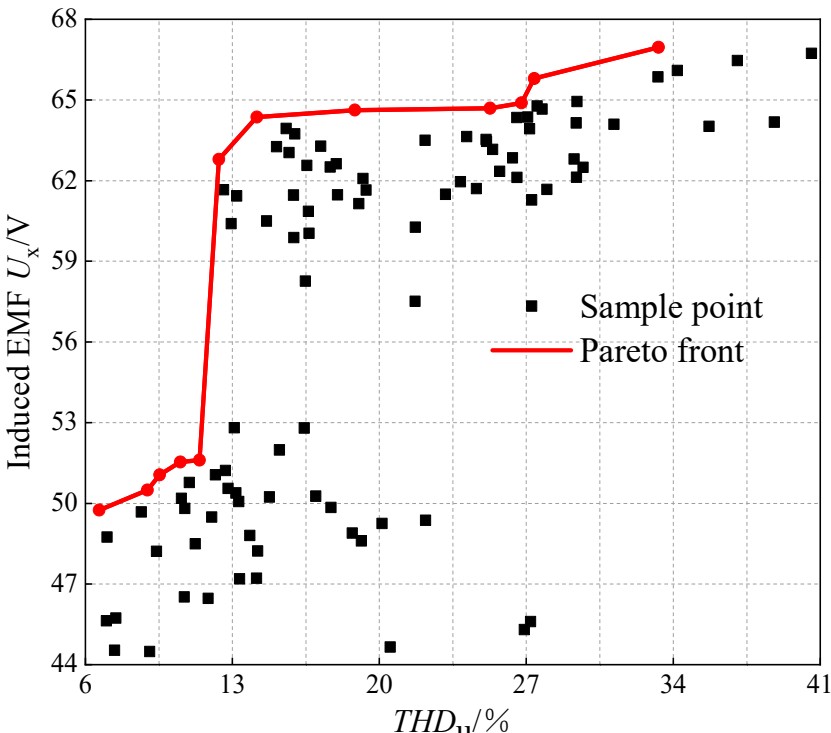

**Figure 13.** Pareto frontier distribution map.

To determine the relatively optimal solution, define the parameter matching coefficient $K_s$, and give weights to the two optimization targets; the larger the value, the better the corresponding generator output performance. The expression is as follows:

$$K_s = C_1 \frac{U_x(x_1, x_2, \ldots, x_m)}{U_0} - \frac{C_2}{100} THD_u(x_1, x_2, \ldots, x_m) \quad (44)$$

where $K_s$ is the performance parameter matching coefficient, which represents the final performance superiority of the generator, $C_1$ and $C_2$ are the weighted coefficients, where $C_1$ takes 0.64 and $C_2$ takes 0.34, $U_0$ is the initial value of the design variable when the generator induced EMF fundamental amplitude.

The relationship between two optimization targets and $K_s$ of the 12 valid solutions is solved by the Pareto frontier solved by this equation. The relationship between the two optimization goals and the matching coefficients is shown in Figure 14.

Figure 14 shows that sample point No. 68 is the "optimal solution" of the optimization model. At this time, the generator induction EMF fundamental amplitude is 42.5 V, the waveform distortion rate is 14.16%, and the final value is obtained according to the comprehensive consideration of each variable value and the actual processing process at this time. The optimal solution and final values of each response variable are shown in Table 4.

**Table 4.** The optimal solution and the final value of each response variable.

| Parameter | Optimal Solution | Final Value |
|:---:|:---:|:---:|
| $k_{cj}$ | 1.1445 | 1.14 |
| $k_{cg}$ | 0.3395 | 0.34 |
| $h_{flp}$/mm | 5.85 | 6 |
| $h_{cj}$/mm | 2.0975 | 2.1 |
| $h_{cg}$/mm | 6.3125 | 6.3 |
| $h_{jyc}$/mm | 4.91 | 3 |

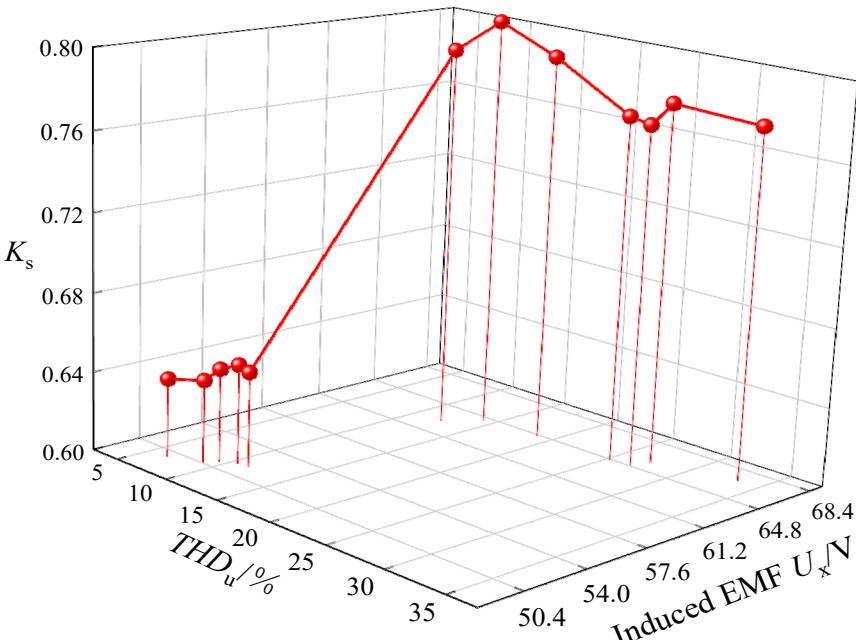

**Figure 14.** Relationship between two optimization goals and matching coefficients.

Among them, the radial thickness of the inter-pole PM $h_{jyc}$ is changed from the optimal value of 4.91 mm to 3 mm, which is due to the low sensitivity of this variable to the two target response values. The change of its value within the constraint range has little impact on the final result. To improve the utilization of the PM, the value of this variable can be selected about the minimum value of the effective solution set.

### 3.2. Multi-Objective Optimization of the Salient-Pole Rotor

The influence of various parameters of the salient-pole rotor on the generator's performance is also complex and changeable. This paper adopts the particle swarm optimization algorithm for the salient-pole rotor for multi-objective optimization design. The particle swarm optimization algorithm (PSO) is derived from the study of the predation behavior of bird flocks, which is essentially a random search algorithm, and as an emerging optimization algorithm, the PSO has a faster computing speed and better global search capabilities than traditional algorithms and is suitable for seeking optimization in a dynamic, multi-objective optimization environment [34,35].

#### 3.2.1. Design Variables and Optimization Goals

After the previous analysis, the magnetization thickness, tangential length, and angle of the V-shaped PM in the salient-pole rotor, and the pole arc coefficient and eccentric distance of the salient-pole shoe will have a greater impact on the air gap magnetic density, so these five parameters are used as design variables, and the air gap magnetic density base wave amplitude and sine waveform distortion rate is optimized. The constraints for each design variable are shown in Table 5.

The optimization goal is that when the five design variables meet the constraints, the amplitude of the air gap magnetic density fundamental wave is the largest and the waveform distortion rate is the smallest, and the optimization model is:

$$\begin{cases} \max B_x(x_1, x_2, \ldots, x_m) \\ \min THD_B(x_1, x_2, \ldots, x_m) \end{cases} \tag{45}$$

where $B_x$ and $THD_B$ are functions for five design variables.

**Table 5.** The constraint scope of the design variable.

| Design Variables | Initial Value | Constraints |
|---|---|---|
| Magnetization thickness of PM $a_{r2}$/mm | 2 | $2 \leq a_{r2} \leq 3$ |
| Tangential length of PM $b_{r2}$/mm | 6 | $5 \leq b_{r2} \leq 7.5$ |
| PM angle $\alpha_{r2}$/° | 55 | $50 \leq \alpha_{r2} \leq 100$ |
| Pole-arc coefficient $k_{r2}$ | 0.85 | $0.8 \leq k_{r2} \leq 0.9$ |
| Eccentricity $h_{px}$/mm | 11 | $5 \leq h_{px} \leq 20$ |

3.2.2. Particle Swarm Algorithm Flow

Assuming that there is a population of $N$ particles in a $D$-dimensional target search space, the position vector and velocity vector of the $i$th particle are defined respectively as follows [36]:

$$
\begin{aligned}
x_i &= (x_{i1}, x_{i2}, \ldots, x_{iD}), i = 1, 2, \ldots, N \\
v_i &= (v_{i1}, v_{i2}, \ldots, v_{iD}), i = 1, 2, \ldots, N
\end{aligned}
\tag{46}
$$

The optimal position found by the $i$th particle is counted as:

$$
p_{\text{best}} = (p_{i1}, p_{i2}, \ldots, p_{iD}), i = 1, 2, \ldots, N
\tag{47}
$$

The global optimal value in the particle swarm can be expressed as:

$$
g_{\text{best}} = (g_{g1}, g_{g2}, \ldots, g_{gD})
\tag{48}
$$

The updated position and speed of the $i$th particle are as follows:

$$
\begin{cases}
x_{id} = x_{id} + v_{id} \\
v_{id} = \omega \cdot v_{id} + c_1 r_1 (p_{id} - x_{id}) + c_2 r_2 \left(p_{gd} - x_{id}\right)
\end{cases}
, d = 1, 2, \ldots, D
\tag{49}
$$

where $\omega$ is the inertia weight coefficient, $c_1$ and $c_2$ are learning factors, and $r_1$ and $r_2$ are random numbers of [0, 1].

In most cases, the optimal goals found in multi-objective optimization are mutually restrictive and conflicting, so the result of multi-objective optimization is not a unique value, but a solution set, that is, a Pareto optimal solution. In the Pareto solution set, each valid solution has advantages and disadvantages, and cannot be directly used as the final optimization result, so it is necessary to select a set of solutions as the final result according to the design intent. In this article, the linear weighting method is chosen, the optimization target is given a weight and then added together to form a single objective function solution, which can be expressed as:

$$
F(x) = k_{bx}(B_x(x) - B_0) - k_{thd}THD_B
\tag{50}
$$

where $k_{bx}$ is the weighted coefficient for the amplitude of the magnetic density fundamental wave of the air gap, $k_{bx}$ = 0.85, $k_{thd}$ is the weighted coefficient for the distortion rate of the air gap magnetic density waveform, $k_{thd}$ = 0.27, $B_0$ is the initial value of the generator air gap magnetic density fundamental amplitude.

The flow block diagram of the particle swarm optimization algorithm is shown in Figure 15 [37], where the particle swarm population size is set to 200, the number of iterations is 300, the learning factor is 0.9, the maximum inertia factor is 0.9, and the minimum inertia factor is 0.4.

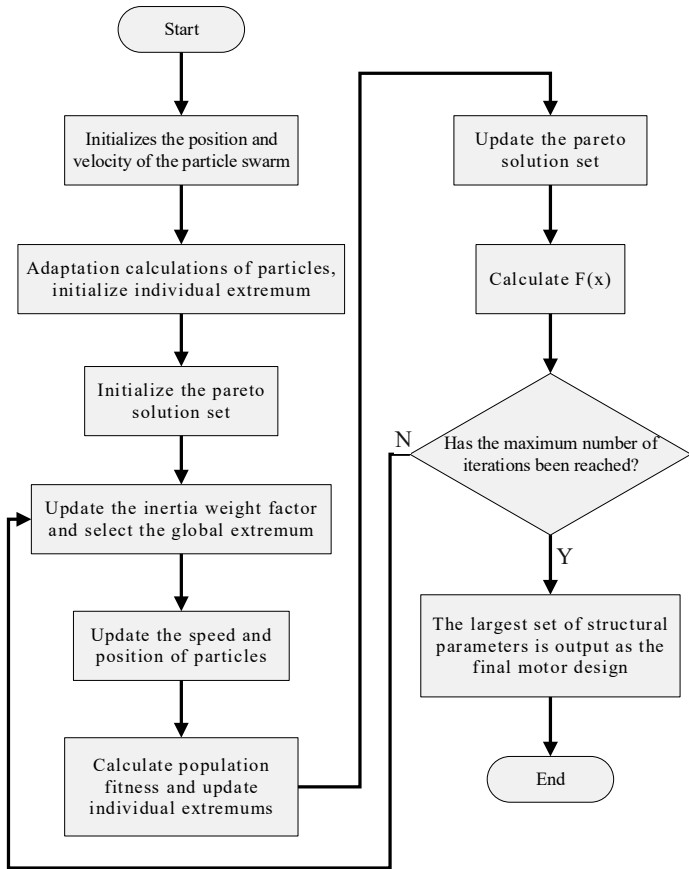

**Figure 15.** Flow block diagram of the particle swarm optimization algorithm.

### 3.2.3. Particle Swarm Algorithm Optimization Results

Figure 16 shows the relationship between the evolutionary algebra and the optimal individual values of each generation, and Table 6 shows the initial and optimal values of the design variables.

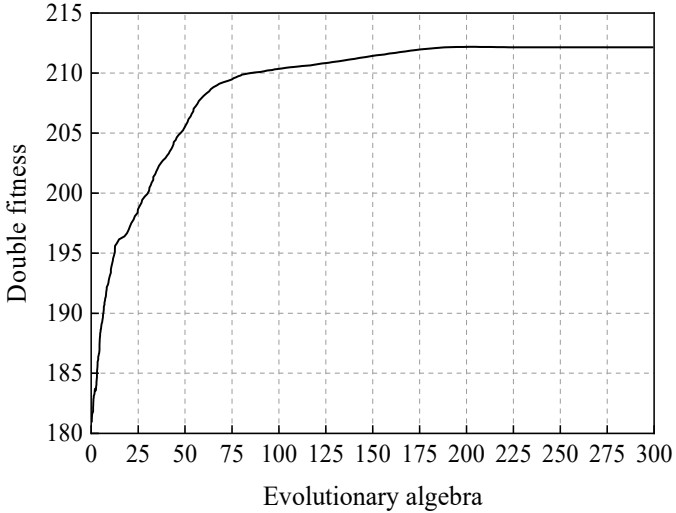

**Figure 16.** Diagram of the relationship between evolutionary algebra and optimal individual values.

**Table 6.** The constraint scope of the design variable.

| Design Variables | Initial Value | Optimal Value |
|---|---|---|
| Magnetization thickness of PM $a_{r2}$/mm | 2 | 2.39129 |
| Tangential length of PM $b_{r2}$/mm | 6 | 6.68317 |
| PM angle $\alpha_{r2}$/° | 55 | 61.6139 |
| Pole-arc coefficient $k_{r2}$ | 0.85 | 0.809997 |
| Eccentricity $h_{px}$/mm | 11 | 12.5815 |

According to the processing process conditions and generator design experience, the final values of the five parameters can be determined as follows: the magnetization thickness of PM $a_{r2}$ takes 2.4 mm, the tangential length of PM $b_{r2}$ takes 6.7 mm, the angle of PM $\alpha_{r2}$ takes 61.6°, the polar arc coefficient $k_{r2}$ takes 0.81, and the eccentric distance $h_{px}$ takes 12.6 mm.

*3.3. Simulation Analysis of Hybrid Excitation Generator*

3.3.1. Simulation Analysis of Optimization Results

After the optimization of the key structural parameters of the claw pole and the optimization of multi-objectives, the output performance of the generator has been greatly improved, and the change of the induction EMF of the generator no-load before and after the optimization is shown in Figure 17.

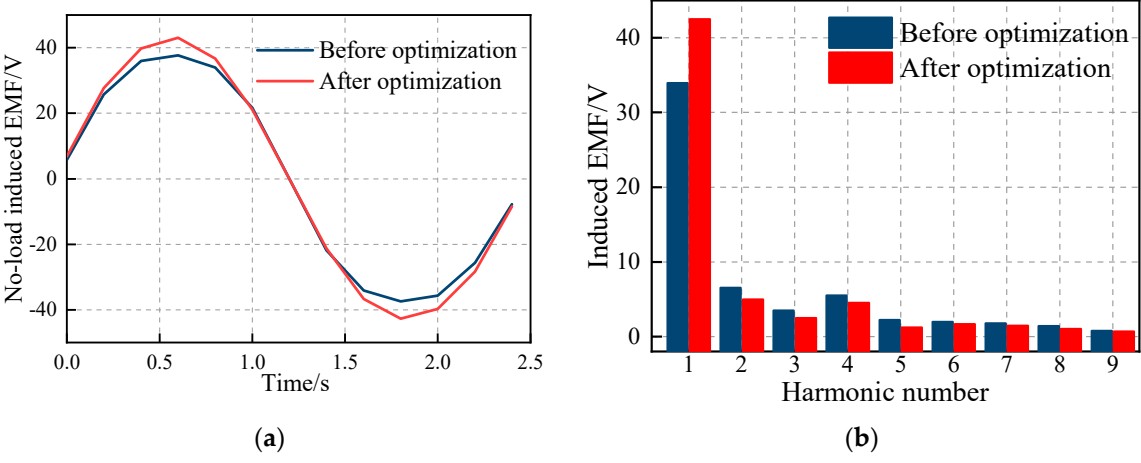

(**a**)　　　　　　　　　　　　　　(**b**)

**Figure 17.** Comparison of the no-load induced EMF of claw pole rotor before and after optimization. (**a**) Comparison of the no-load induced EMF curves; (**b**) comparison of the no-load induced EMF harmonic distribution.

From Figure 17, it can be seen that after the improvement and optimization of the claw pole generator structure, the amplitude of the no-load induced EMF fundamental wave of the generator has been greatly improved from 34.5 V to 42.5 V, and the amplitude of each harmonic has decreased, and the performance of the generator has been further improved. The output voltage and cogging torque change of the generator before and after the optimization are shown in Figures 18 and 19.

It can be seen from Figures 18 and 19 that the output voltage of the generator increased significantly, and the average value increased from 59.2 V to 67.3 V, with an increase of 13.6%. Although the cogging torque also increased slightly, the peak increased from 0.179 N·m to 0.22 N·m, with an increase of 23%. Due to the advantages of the small cogging torque of the claw pole generator, it is still in the acceptable range. The Latin hypercube sampling-Pareto frontier solution method optimizes the structural parameters of the claw pole rotor.

At this time, the performance of the generator has been improved compared with the prototype, and the air gap magnetic density before and after the optimization is shown in Figure 20.

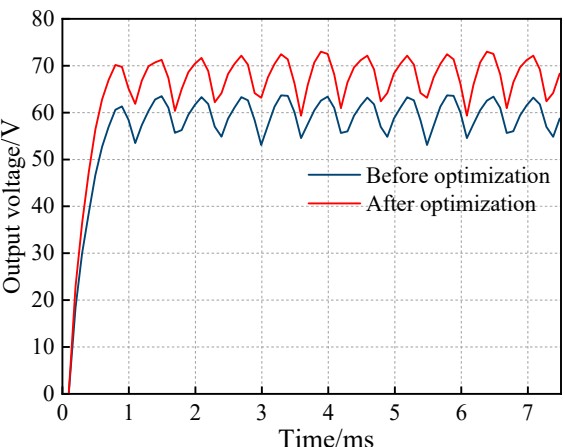

**Figure 18.** Comparison of the output voltage of the claw pole rotor before and after optimization.

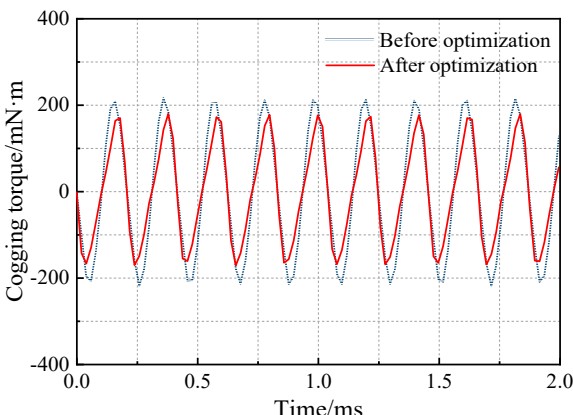

**Figure 19.** Comparison of cogging torque of claw pole rotor before and after optimization.

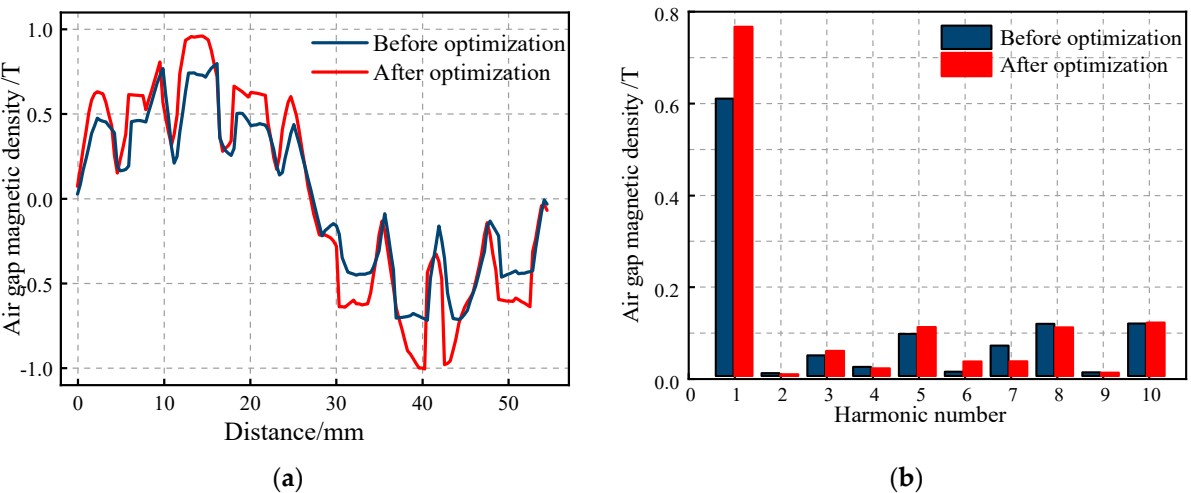

**Figure 20.** Comparison of the air gap magnetic flux density of the salient-pole rotor before and after optimization. (**a**) Comparison of the air gap magnetic flux density curves; (**b**) comparison of the air gap magnetic flux density harmonic distribution.

From Figure 20, it can be seen that the peak of the air gap magnetic flux density of the generator after optimization has been significantly improved, and the fundamental amplitude increased from 0.89 T to 0.93 T, which is an increase of 4.5%. The amplitude of the 2nd, 3rd, 4th, 5th, and 7th harmonics has been significantly reduced, and the distortion rate of the air gap magnetic flux density waveform has decreased from 39.6% to 18.6%. The waveform of air gap flux density is closer to sine, which improves the output performance of the generator. The comparison of no-load induced EMF of the generator before and after optimization is shown in Figure 21.

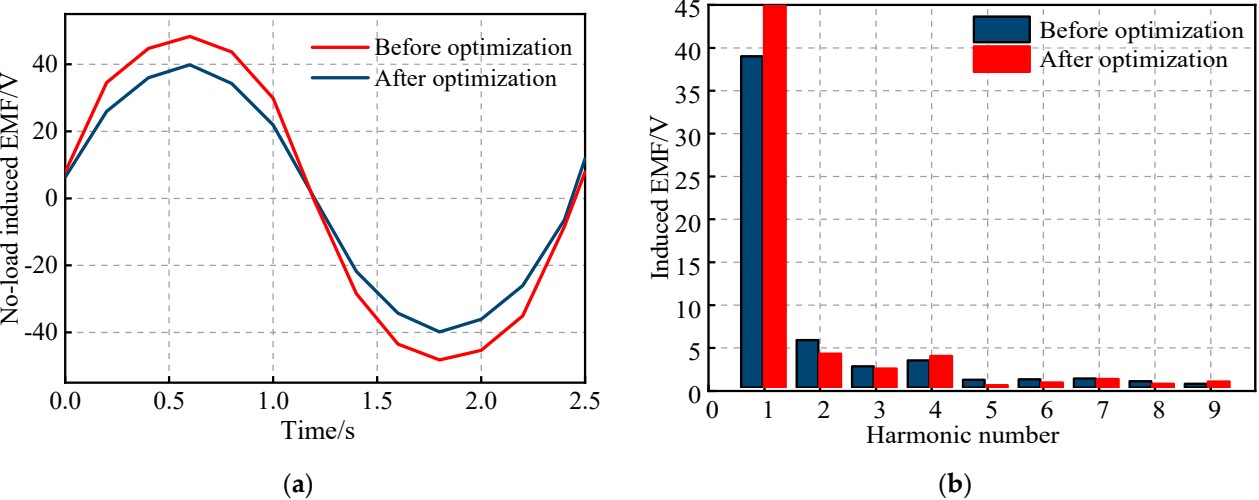

(**a**)                    (**b**)

**Figure 21.** Comparison of induced EMF of the generator before and after optimization. (**a**) Comparison of induced EMF curves; (**b**) comparison of induced EMF harmonic distribution.

From Figure 21, it can be seen that the optimized generator no-load induction EMF waveform is more sinusoidal, and the peak value is increased from 39.5 V to 47.9 V, an increase of 15.2%. The induced EMF fundamental wave amplitude is increased from 38.6 V to 44.8 V, of which the amplitude of the second harmonic is greatly reduced, and the distortion rate of the sine waveform is reduced. In the case of the same volume of the generator, by optimizing the structural parameters, the output voltage of the generator is increased.

### 3.3.2. Loss Simulation Analysis

The no-load core loss is mainly distributed in the stator teeth and the stator yoke, and the core loss of the rotor is mainly located on the rotor pole surface. It can be seen from Figure 22 that the average value of no-load core loss is about 14 W, and the average value of rated iron loss is about 19 W. Due to the increase of load, the core loss under rated state is slightly higher than that under no-load. The copper loss output of the hybrid magnetic circuit generator is relatively stable under the rated load, about 140 W, which meets the loss requirements when the generator operates stably.

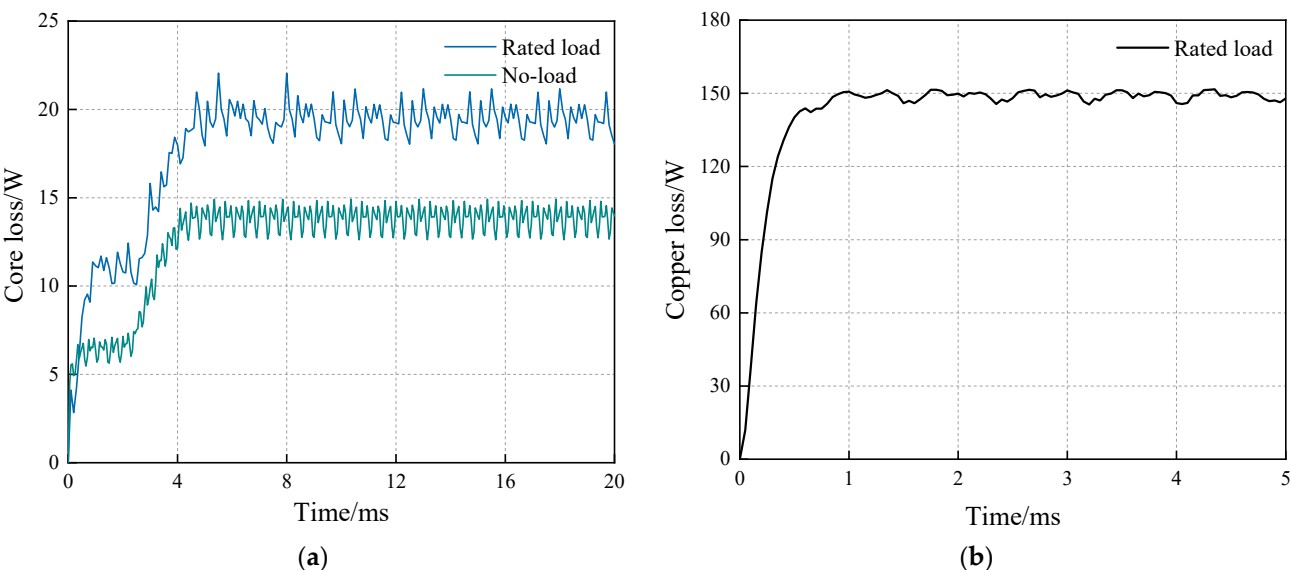

(**a**)                                    (**b**)

**Figure 22.** Loss of the hybrid excitation generator. (**a**) Core loss of the generator; (**b**) copper loss of the generator.

## 4. Experimental Validation

According to the optimization design parameters, the prototype of the compound excitation generator is developed, the experimental platform is built, and the prototype experiment is carried out. The claw pole rotor of the compound excitation generator is juxtaposed with the salient-pole rotor, sharing a set of armature windings. The whole generator has a compact structure, and there is a magnetic isolation bushing in the middle of the two rotors, which can effectively avoid axial magnetic leakage. The prototype developed according to the structure of the compound excitation generator is shown in Figure 23.

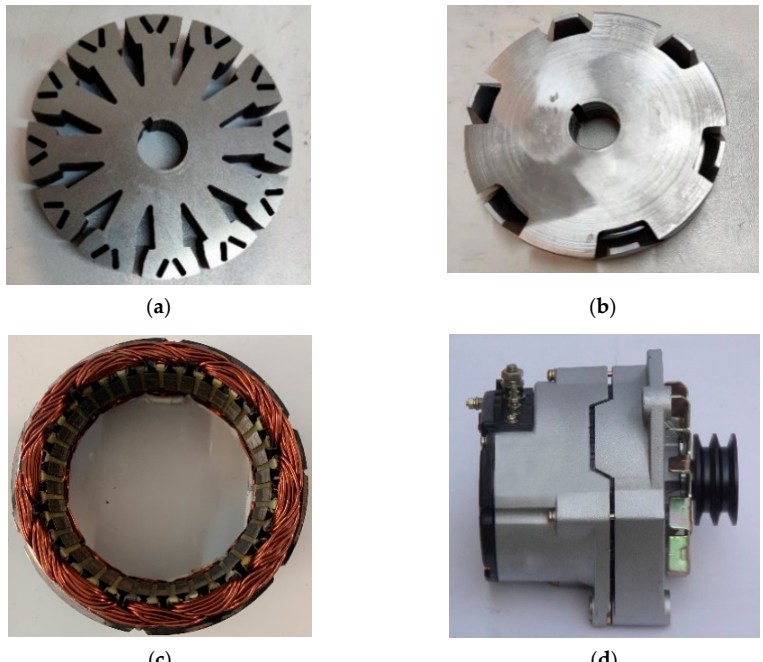

**Figure 23.** Compound excitation generator prototype. (**a**) Salient-pole rotor core; (**b**) PM claw pole rotor core; (**c**) stator; (**d**) generator.

The prototype experimental platform of the compound excitation generator is shown in Figure 24. This experiment mainly uses manual adjustment of excitation current to control the generator, the experimental results are output by the oscilloscope.

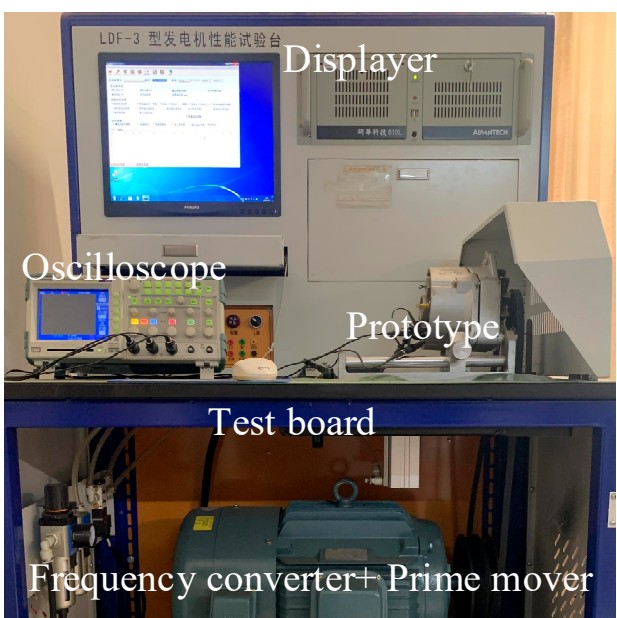

**Figure 24.** Compound excitation generator prototype experimental platform.

### 4.1. Generator Output Characteristics

The generator prototype is externally connected with pure resistive load and the generator is set to run at 4000 r/min with the excitation current 2 A. The induced EMF of the generator when measuring the size of different load resistances is shown in Figure 25.

As can be seen from Figure 25, with the reduction of the resistive load resistance, the current through the resistive load gradually increases, and the current in the corresponding armature winding also gradually increases. At this time, the armature reaction of the generator will gradually increase, the armature reaction will have a more serious interference with the atmospheric magnetic field, the induced EMF sinusoidally generated by the armature winding is rapidly reduced, and the amplitude is gradually reduced. At this time, the output voltage of the generator will be reduced. If you want to maintain a certain voltage value, you need to increase the excitation current. After rectification, the relation diagram of the load current, the output voltage and output power can be obtained, and compared with the simulation data, the load characteristic curve of the compound excitation generator is obtained as shown in Figure 26.

As can be seen from Figure 26, when the rated speed and excitation current of the generator remain unchanged, with the increase of the load, the load current continues to increase, and the output voltage of the generator gradually decreases, the output voltage of the generator is 82 V when no-load, the load current under the rated load is 35.7 A, and the output voltage is 28 V, which has met the design requirements.

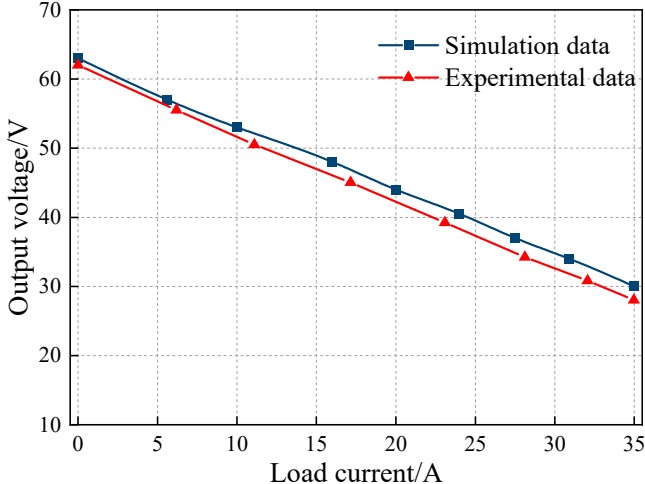

**Figure 25.** Induced EMF of the compound excitation generator prototype under different loads. (**a**) Load 10 Ω; (**b**) Load 5 Ω; (**c**) Load 3 Ω; (**d**) Load 1.5 Ω.

**Figure 26.** Compound excitation generator external characteristic curve.

*4.2. Generator Output Power and Efficiency*

After rectification, the relation between load current and output power is obtained when the generator speed is 4000 r/min, and compared with simulation data, the power

curve of the hybrid excitation generator shown in Figure 27a is obtained. From the diagram, it can be seen that output power of generator increases with the increase of current, and the experimental data is slightly lower than simulation data, and the output power is 982 W under rated load. Figure 27b shows the variation curve of power and efficiency of the generator with the increase of speed. As shown in Figure 27b, the power and efficiency of the generator increases with speed. In the working range, the efficiency of the generator is stable at over 90%, which meets the design requirements.

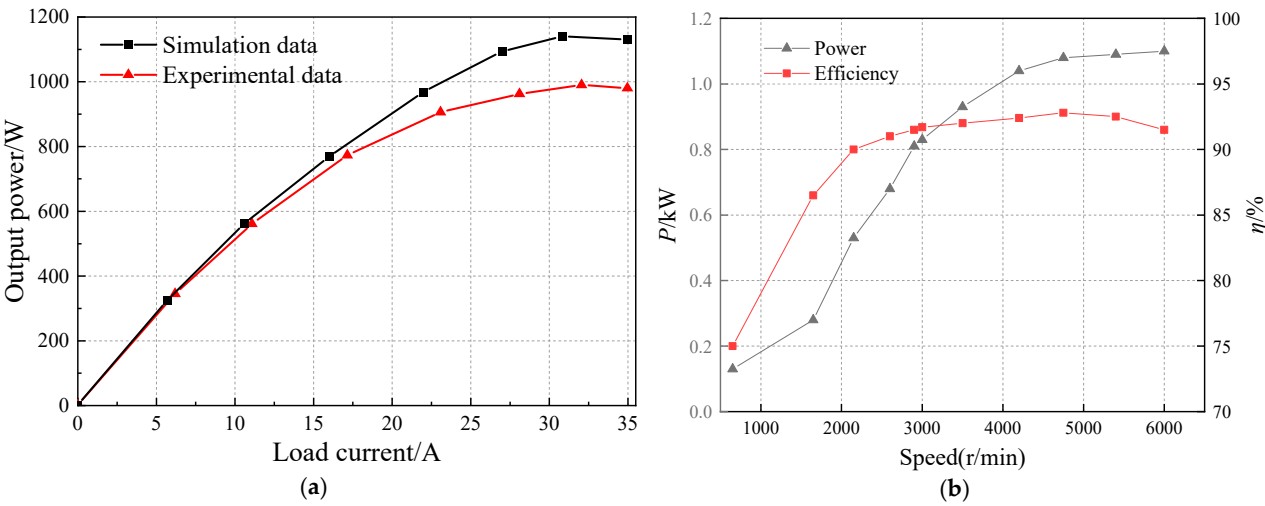

(**a**)                                    (**b**)

**Figure 27.** Generator output power and efficiency change curve. (**a**) The output power varies with the load current; (**b**) curve of power and efficiency with generator speed.

### 4.3. Generator Regulation Characteristic

The regulation ability of the compound excitation generator makes it have a relatively simple control system, so the regulation characteristics of the generator should be tested experimentally. The generator is run at a rated speed of 4000 r/min, by changing the size of the load resistance to change the load current, so that the generator output voltage is always stable at 28 V, measuring the size of the excitation current under different load currents, and obtain the adjustment characteristic curve of the compound excitation generator prototype as shown in Figure 28.

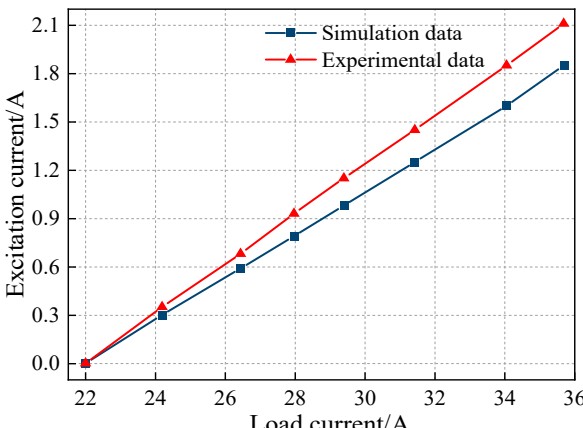

**Figure 28.** The compound excitation generator regulates the characteristic curve.

As can be seen from Figure 28, when the generator load is 616 W, the excitation current is 0 A, and the compound excitation generator output voltage can reach 28 V without the need for an electrical excitation magnetic field. When the generator is rated at 1000 W, the excitation current is 2.1 A, and the increase of the electrical excitation magnetic field makes

the generator output voltage still reach 28 V. It can be seen that the introduction of the electric excitation magnetic field can offset the voltage reduction caused by the increase in the load of the generator, improve the strength of the generator magnetic field, and make the generator output voltage always stable at the target voltage value.

When the load power is 980 W, 1000 W, and 1020 W, the performance experiment of the compound excitation generator from low speed to high speed is carried out, and the experimental results are shown in Table 7.

**Table 7.** Compound excitation generator output performance experimental results.

| Speed (r/min) | Load Power (W) | Output Voltage (V) |
| --- | --- | --- |
| 2000 | 980 | 27.6 |
| | 1000 | 27.1 |
| | 1020 | 26.5 |
| 4000 | 980 | 28.3 |
| | 1000 | 28.4 |
| | 1020 | 28.2 |
| 4800 | 980 | 28.6 |
| | 1000 | 28.6 |
| | 1020 | 28.5 |

As can be seen from Table 7, when the generator speed changes from 2000 r/min to 4800 r/min and the load power changes from 980 W to 1020 W, the output voltage is always stable between 26.5 and 28.6 V. It can be seen that by adjusting the size of the excitation current and the high-speed magnetic weakening control of the three-phase half-control bridge rectifier voltage regulation circuit, the compound excitation generator can output a stable rated voltage at different speeds and loads.

## 5. Conclusions

A new type of dual-rotor compound excitation generator structure is proposed so that the generator has a PM claw pole rotor and a parallel magnetic circuit salient-pole rotor, the two are parallel coaxial, sharing an armature winding, and the generated magnetic field is synthesized in the air gap. To further improve the performance of the designed double-rotor compound excitation generator, the rotor structure is optimized. We get the following conclusions:

- For the PM claw rotor containing the inter-pole PM, the induced EMF is analyzed and calculated in the form of the equivalent magnetic circuit method and the improved equivalent magnetic network method, which improves the calculation accuracy and ensures the operation speed, and then the structural parameters of the PM claw rotor are optimized based on Latin hypercube sampling, and the optimal parameter matching of the claw rotor is determined.
- For the salient-pole rotor, the air gap magnetic density is calculated by the improved subdomain method, the salient-pole rotor with uneven excitation source is analyzed and calculated by the equivalent magnetic field and boundary method, and the structural parameters of the salient-pole rotor are optimized based on the particle swarm algorithm, and the optimal parameter matching of the salient-pole rotor is determined.
- Through the comparative analysis of the performance of the generator before and after the optimization of the rotor structure, it can be seen that the optimization of the claw rotor increased the amplitude of the generator induction EMF fundamental wave from 39.2 V to 43.1 V, an increase of 10.2%, and the optimization of the salient-pole rotor increased the generator induction EMF fundamental amplitude from 38.6 V to 44.8 V, an increase of 16%.
- The above optimized design can greatly improve the performance of the compound excitation generator. The final prototype experiment shows that under the conditions

of variable speed and variable load, the output voltage of the new hybrid excitation generator developed can be stabilized between 26.5 V and 28.6 V, and the generator has better output performance and higher power density.

The magnetic field analysis calculation method applied in this paper simplifies the complex analytical model and has reference significance for the establishment of the mathematical model of the magnetic field of this type of generator. In addition, the two multi-objective optimization methods used for the double rotor of the generator in this paper are simple and practical and have certain practical application value for the design and optimization of high-performance hybrid excitation generators.

**Author Contributions:** Conceptualization, X.Z. and S.Y.; methodology, S.Y.; software, S.Y.; validation, S.Y., M.X. and Y.Z.; formal analysis, S.H.; investigation, S.Y.; resources, X.Z.; data curation, Y.Z.; writing—original draft preparation, S.Y.; writing—review and editing, T.G.; visualization, S.Y.; supervision, J.Z.; project administration, M.X.; funding acquisition, X.Z. All authors have read and agreed to the published version of the manuscript.

**Funding:** This research was funded by the National Natural Science Foundation of China (Grant nos. 51875327 and 51975340).

**Institutional Review Board Statement:** Not applicable.

**Informed Consent Statement:** Not applicable.

**Data Availability Statement:** Not applicable.

**Conflicts of Interest:** The authors declare no conflict of interest.

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
