# Peer review of "Magnetic Field Analysis and Performance Optimization of Dual-Rotor Hybrid Excitation Generator for Automobile"

_machines, doi:10.3390/machines10090816_

Round 1

Reviewer 1 Report

The paper proposes a hybrid excitation generator, which adopts a salient-pole rotor and claw-pole rotor. The presentation, organization, and technical illustration of the paper are satisfactory. I think it can be directly accepted for publication after modification. Here are some of my suggestions:

(1) The PM steel in the salient-pole rotor is designed as a "V" shape. What are the advantages compared with other structures? I think these advantages should be described.

(2) The design parameters kcg and hcg in Table 1 are inconsistent with those in the paper and need to be unified.

(3) The IEMF in Fig. 16 (a) and Fig. 20 (a) should be unified.

(4) In the experiment part, only four load conditions are tested. Are the four load conditions set enough? Should more conditions be added?

(5) The English improvement including grammatical errors like comma, space & paragraph settings is required.

Reviewer 2 Report

The article theoretically discusses the optimal design of a synchronous generator with hybrid excitation. A limited experimental verification is provided. The article may be of interest to researchers in the field, but it is necessary to explain some points:

1) For an electric generator, the most important characteristics include efficiency and power losses. Please provide a detailed description of the method for calculating losses in the proposed generator. Also provide results of the calculation of the separate generator losses in various loading points.

2) Please provide a comparison of the calculated separate losses (in copper, in steel, etc.) and with the experimental data.

Reviewer 3 Report

1) The contributions of the paper are not clear.

2) Please add the organization/structure of the paper in the end of the introduction section.

3) What is novelty in the mathematics given in equations 1-8 ? similarly for the mathematics given in other sections of the paper.

4) On page 10, few assumptions are taken to simplify the model, what are the benefits and drawbacks of using these assumptions in the model? 

5) For multiobjective optimization, what is the cost function ?
